# EFFECTIVE AND PARAMETER-EFFICIENT REUSING FINE-TUNED MODELS

## ABSTRACT

Many pre-trained large-scale models provided online have become highly effective in transferring to downstream tasks. At the same time, various task-specific models fine-tuned on these pre-trained models are available online for public use. In practice, as collecting task-specific data is labor-intensive and fine-tuning the large pre-trained models is computationally expensive, one can reuse task-specific fine-tuned models to deal with downstream tasks. However, using a model per task causes a heavy burden on storage and serving. Recently, many training-free and parameter-efficient methods have been proposed for reusing multiple fine-tuned task-specific models into a single multi-task model. However, these methods exhibit a large accuracy gap compared with using a fine-tuned model per task. In this paper, we propose **P**arameter-**E**fficient methods for **R**e**U**sing (PERU) fine-tuned models. For reusing **F**ully **F**ine-**T**uned (FFT) models, we propose PERU-FFT by injecting sparse task vectors into a merged model by magnitude pruning. For reusing LoRA fine-tuned models, we propose PERU-LoRA use a lower-rank matrix to approximate the LoRA matrix by singular value decomposition. Both PERU-FFT and PERU-LoRA are training-free. Extensive experiments conducted on computer vision and natural language process tasks demonstrate the effectiveness and parameter-efficiency of the proposed methods. The proposed PERU-FFT and PERU-LoRA outperform existing merging models method by a large margin and achieve comparable performance to using a fine-tuned model per task. PERU-FFT is general and can be integrated into any existing merging models methods to boost performance.

## 1 INTRODUCTION

In recent years, large-scale models pre-trained on massive data have proven effective in transferring to downstream tasks (Chen et al., 2022; Min et al., 2022; Yuan et al., 2023; Ruiz et al., 2023). Various pre-trained models are available on Hugging Face (Wolf et al., 2020), e.g., *ResNet* (He et al., 2016), *ViT* (Dosovitskiy et al., 2021), *CLIP* (Radford et al., 2021), and diffusion models (Ho et al., 2020; Rombach et al., 2022) for computer vision; *T5* (Raffel et al., 2020), *GPT-2* (Radford et al., 2019), and *LLaMA* (Touvron et al., 2023a;b) models for natural language processing. Practitioners specialize a pre-trained model to a task-specific model by either fully or parameter-efficient fine-tuning (Houlsby et al., 2019; Hu et al., 2022; Lester et al., 2021; Jiang et al., 2023; Yu et al., 2023) on the task data, e.g., a *CLIP-L/14* model (Radford et al., 2021) fine-tuned on the *SUN397* benchmark (Xiao et al., 2016) can be used for scene recognition tasks. Many fine-tuned models are published online for public use. By 2023, more than $120,000$ models are available on Hugging Face Hub.

For a downstream task, as collecting task-specific data is labor-intensive and fine-tuning the large pre-trained models is computationally expensive, one can download and reuse the fine-tuned models from Hugging Face. In real-world applications, we usually need to deal with a number of tasks simultaneously (Dong et al., 2015; Siam et al., 2018; Raffel et al., 2020). Using a task-specific fine-tuned model for each task is effective but costly in storing and serving. Fine-tuning the pre-trained model on all task data can address this issue but requires expensive re-training and the availability of all task data, which is infeasible.

Recently, many training-free and parameter-efficient methods have been proposed for merging multiple fine-tuned task-specific models into a single multi-task model. For example, Task-Arithmetic (Ilharco et al., 2023) performs a uniformly merging by adding the average of all task vectors (i.e.,

the difference between the task model and the pre-trained model) to the pre-trained model, while Fisher-Merging (Matena & Raffel, 2022) improves uniform merging to weighted merging, where the weight for each task model is determined by Fisher information matrix estimated on the validation loss. RegMean (Jin et al., 2023) further proposes to merge linear layers by solving a local linear regression problem. TIES-Merging (Yadav et al., 2023) trims low-magnitude elements in the task vectors and attempts to resolve sign disagreements across task models before merging models. For complex tasks, merging task models into a shared model may cause parameter inference (Yadav et al., 2023). Figure 1 shows the average testing accuracy on eight tasks when reusing fine-tuned *ViT* models (Dosovitskiy et al., 2021; Radford et al., 2021), demonstrating a large gap between the accuracy of existing merging methods (Task-Arithmetic, Fisher-Merging, RegMean, TIES-Merging) and using single-task fine-tuned models (denoted Single-Task).

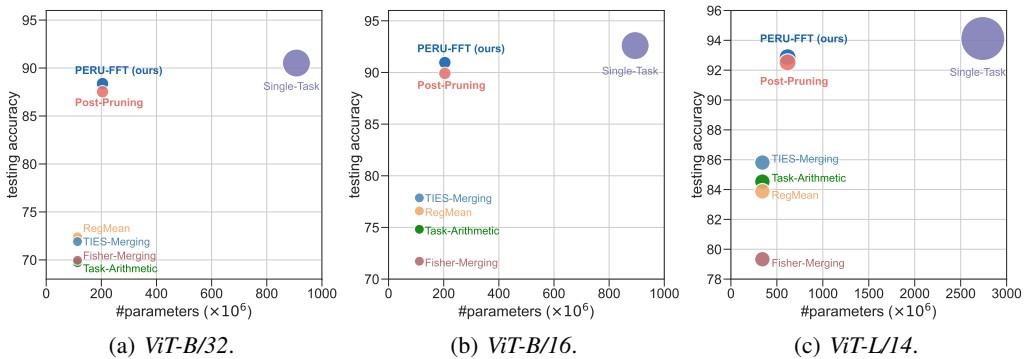

(a) *ViT-B/32.*        (b) *ViT-B/16.*        (c) *ViT-L/14.*

Figure 1: Average testing accuracy on eight tasks by reusing fully fine-tuned models. Post-Pruning and PERU-FFT keeps top-$10\%$ values.

In this paper, we propose PERU, a **P**arameter-**E**fficient method for **ReU**sing fine-tuned models. We first introduce post-tuning technique (Zhu & Gupta, 2017; Liu et al., 2018; Wang et al., 2020; Zhang et al., 2022; Xia et al., 2022) to extract a sparse task vector. This method is simple and effective (Figure 1, Post-Pruning keeps top-10% values). We further propose PERU-FFT to extract task-shared knowledge by merging task-specific models, and prune the difference between the task-specific model and the merged model to extract a sparse vector containing task-specific knowledge. As shown in Figure 1, With only top-$10\%$ values of task vectors, PERU-FFT achieves comparable performance with Single-Task and moreover, performs better than existing merging algorithms by a large margin.

For LoRA Fine-tuned models, the sparsifying task vectors method is not suitable as pruning the LoRA matrices leads to worse performance while pruning their product cannot reduce the number of parameters compared with the LoRA matrices. To address this problem, we propose PERU-LoRA to approximate LoRA matrices by lower rank-$q$ matrices by singular value decomposition. We only need to keep the top-$q$ singular values and their corresponding singular vectors. Empirically, the approximate error decreases exponentially fast w.r.t. $q$, while the accuracy increases exponentially fast. Particularly, PERU-LoRA methods with $q = 16$ achieve comparable performance compared with Single-Task (Figure 2).

Our contributions are summarized as follows: (i) We propose PERU-FFT for reusing fully fine-tuned models, where task vectors are computed from the merged model and the task-specific models. (ii) We propose PERU-LoRA for reusing LoRA fine-tuned models, where the lower-rank matrices are added to $\boldsymbol{\theta}_0$. (iii) Extensive experimental results on computer vision and natural language processing tasks, show that PERU-FFT and PERU-LoRA perform much better than existing merging methods. Furthermore, PERU-FFT and PERU-LoRA achieve comparable performance compared to Single-Task fine-tuned models, but are much more parameter-efficient. (iv) PERU-FFT is general and can be combined with any existing merging algorithms (e.g., Task-Arithmetic (Ilharco et al., 2023), Fisher-Merging (Matena & Raffel, 2022), RegMean (Jin et al., 2023), TIES-Merging (Yadav et al., 2023)) to boost performance.

## 2    RELATED WORKS

We consider a neural network $f(\mathbf{x}; \boldsymbol{\theta})$ with input $\mathbf{x}$ and parameters $\boldsymbol{\theta} \in \mathbb{R}^d$. Let $\boldsymbol{\theta}_0$ be a pre-trained model provided on torchvision (Marcel & Rodriguez, 2010), HuggingFace (Wolf et al., 2020),

or timm (Wightman, 2019), e.g., *ViT-B/32* (Dosovitskiy et al., 2021). Besides, many task-specific models fine-tuned from $\boldsymbol{\theta}_0$ are also publicly available online.

Given $T$ tasks, where each task has a fine-tuned model. We aim to reuse existing fine-tuned models $\{\boldsymbol{\theta}_t : t = 1, \ldots, T\}$ to construct a model that can be used for solving $T$ tasks simultaneously. Different from multi-task learning (Kendall et al., 2018; Liu et al., 2021; 2019; Ye et al., 2021; Lin et al., 2022; 2023), training data for all tasks are unavailable. Hence, we cannot learn a multi-task model by jointly re-training on data. Existing methods focus on merging all task-specific models into a model and expect the merged model to have promising performance on all tasks. For example, Task-Arithmetic (Ilharco et al., 2023) merges all model weights as $\boldsymbol{\theta}^\star = \boldsymbol{\theta}_0 + \lambda \sum_{t=1}^{T}(\boldsymbol{\theta}_t - \boldsymbol{\theta}_0)$, where $\lambda$ is a hyperparameter chosen on a small validation set, and $\mathbf{v}_t \equiv \boldsymbol{\theta}_t - \boldsymbol{\theta}_0$ is a task vector represents the element-wise difference between $\boldsymbol{\theta}_t$ and $\boldsymbol{\theta}_0$. When $\lambda = \frac{1}{T}$, $\boldsymbol{\theta}^\star$ becomes uniform averaging all model weights, i.e., the Model soups method in Wortsman et al. (2022a). Wortsman et al. (2022b) ensemble the pre-trained model $\boldsymbol{\theta}_0$ and fine-tuned model $\boldsymbol{\theta}_t$ to improve the robustness of $\boldsymbol{\theta}_t$. Fisher-Merging (Matena & Raffel, 2022) improves uniform merging to weighted merging, where the weights are determined by the Fisher information matrix estimated on the validation set. RegMean (Jin et al., 2023) proposes to merge linear layers by solving a local linear regression problem while merging other layers by uniform averaging. TIES-Merging (Yadav et al., 2023) trims low-magnitude elements in the task vector $\mathbf{v}_t$ and resolves sign disagreements across task models before performing merging models. Ortiz-Jimenez et al. (2023) study how to fine-tune $\boldsymbol{\theta}_0$ on $\mathcal{D}_t$ such that Task-Arithmetic can perform well.

Pruning, which aims to reduce the model size and maintain the model performance, is a popular technique for compressing and sparsifying neural networks. Many pruning methods (Zhu & Gupta, 2017; Liu et al., 2018; Wang et al., 2020; Zhang et al., 2022; Xia et al., 2022) sparse model weights in an optimization or evolutionary manner and need enough training data, gradient information, and even re-training, which is unsuitable for the reusing model problem. For example, Zhang et al. (2022) formulate pruning as a bi-level optimization problem and iteratively optimize to find a binary mask to select model weights. Magnitude-based pruning (Han et al., 2015; Narang et al., 2016; Zhu & Gupta, 2017), which selects weights for a trained model based on the weight magnitudes, is data-free and training-free pruning.

## 3 PARAMETER-EFFICIENT REUSING FINE-TUNED MODELS

### 3.1 REUSING FULLY FINE-TUNED MODELS

For reusing task-specific fine-tuned models, existing methods (e.g., Task-Arithmetic (Ilharco et al., 2023), Fisher-Merging (Matena & Raffel, 2022), RegMean (Jin et al., 2023), TIES-Merging (Yadav et al., 2023)) focus on merging all task models into a shared model without any task-specific parameters. As can be seen from Figure 1, their accuracies (averaged over eight tasks) are much lower than that of Single-Task. To deal with this issue, we propose to inject *sparse* task-specific vectors into the merged model.

In reusing fine-tuned models, training-based pruning methods (Zhu & Gupta, 2017; Liu et al., 2018; Wang et al., 2020; Zhang et al., 2022; Xia et al., 2022) based on weights importance are infeasible for sparsifying the task vectors, since the training data are unavailable. We introduce post-pruning (Han et al., 2015; Narang et al., 2016; Zhu & Gupta, 2017) extracts sparse task-specific vectors from task vectors based on their magnitudes. Compared with training-based pruning, Post-Pruning is training-free. For each task, we keep the top-$m\%$ (e.g., 1%, 10%) values of the task vectors and prune the rests:

$$\hat{\mathbf{v}}_t(m) = \text{keep top-}m\% \text{ of } \mathbf{v}_t \text{ based on magnitude.} \tag{1}$$

In inference, $\boldsymbol{\theta}_0 + \hat{\mathbf{v}}_t(m)$ is used as a pruned task model. The procedure of Post-Pruning is shown in Algorithm 1.

As $\boldsymbol{\theta}_0 + \hat{\mathbf{v}}_t(m)$ only depends on the $t$th task model, it does not use shared knowledge from other tasks. We propose to merge task-specific models before pruning. Specifically, let $\mathbf{u}_t \equiv \boldsymbol{\theta}_t - \boldsymbol{\theta}^\star$, $t = 1, \ldots, T$, where $\boldsymbol{\theta}^\star$ is a merged model. We prune $\mathbf{u}_t$ to $\hat{\mathbf{u}}_t(m)$ by keeping the top-$m\%$ values of $\mathbf{u}_t$ as in (1). In inference, $\boldsymbol{\theta}^\star + \hat{\mathbf{u}}_t(m)$ is used as a pruned task model. As the method for obtaining $\boldsymbol{\theta}^\star$ is flexible, any merging algorithms (e.g., Task-Arithmetic, Fisher-Merging, RegMean, TIES-

Merging) can be adopted. The procedure, called PERU-FFT, is shown in Algorithm 1. Compared with Post-Pruning, PERU-FFT has the same number of parameters for a specific ratio $m\%$.

---

**Algorithm 1** Post-Pruning (resp. PERU-FFT).

**Require:** $m\%$; $\boldsymbol{\theta}_0$; $\boldsymbol{\theta}_1, \ldots, \boldsymbol{\theta}_T$; a merging algorithm $\mathcal{A}_{\text{merging}}$;
  1: if PERU-FFT: obtain $\boldsymbol{\theta}^\star$ by $\mathcal{A}_{\text{merging}}$;
  2: **for** $t = 1, \ldots, T$ **do**
  3:     $\mathbf{v}_t = \boldsymbol{\theta}_t - \boldsymbol{\theta}_0$ (resp. $\mathbf{u}_t = \boldsymbol{\theta}_t - \boldsymbol{\theta}^\star$);
  4:     obtain $\hat{\mathbf{v}}_t(m)$ (resp. $\hat{\mathbf{u}}_t(m)$) by keeping top-$m\%$ values;
  5:     evaluate $\boldsymbol{\theta}_0 + \hat{\mathbf{v}}_t(m)$ (resp. $\boldsymbol{\theta}^\star + \hat{\mathbf{u}}_t(m)$) on task $t$'s testing set;
  6: **end for**

---

### 3.2 REUSING LORA FINE-TUNED MODELS

As pre-trained models are usually huge (e.g., *ViT-L/14* (Dosovitskiy et al., 2021) has 343M parameters, *T5-base* (Raffel et al., 2020) has 220M parameters, *LLaMA-2* (Touvron et al., 2023b) series have 7B, 13B, 70B parameters), LoRA Fine-Tuning (Hu et al., 2022) is a parameter-efficient method to obtain task-specific models. The fine-tuned task model $\boldsymbol{\theta}_t \in \mathbb{R}^{d_{\text{out}} \times d_{\text{in}}}$ is decomposed as

$$\boldsymbol{\theta}_t = \boldsymbol{\theta}_0 + \mathbf{A}_t \mathbf{B}_t^\top, \tag{2}$$

where $\mathbf{A}_t \in \mathbb{R}^{d_{\text{out}} \times r}$, $\mathbf{B}_t \in \mathbb{R}^{d_{\text{in}} \times r}$, and $r \ll \{d_{\text{in}}, d_{\text{out}}\}$. The number of parameters required in LoRA fine-tuning is $r \times (d_{\text{out}} + d_{\text{in}})$, much smaller than that fully fine-tuning ($d_{\text{out}} \times d_{\text{in}}$) as $r$ is usually small, e.g., $r = 128$. Due to its efficiency, many task-specific LoRA fine-tuned models are available online for public use.

Existing methods for merging fully fine-tuned models can be applied directly to merging LoRA fine-tuned models $\{\boldsymbol{\theta}_t : t = 1, \ldots, T\}$. As shown in Figure 2[1], existing methods perform much worse than the Single-Task (LoRA fine-tuned) method. Hence, using a merged model for all tasks without task-specific parameters is undesirable. Different from reusing fully fine-tuned models, sparsifying $\mathbf{A}_t \mathbf{B}_t^\top$ is not parameter-efficient compared with storing $\mathbf{A}_t$ and $\mathbf{B}_t$ separately. In the following, we use singular value decomposition (SVD) to extract a small fraction of parameters from the task-specific LoRA matrix, which is then injected into the shared model.

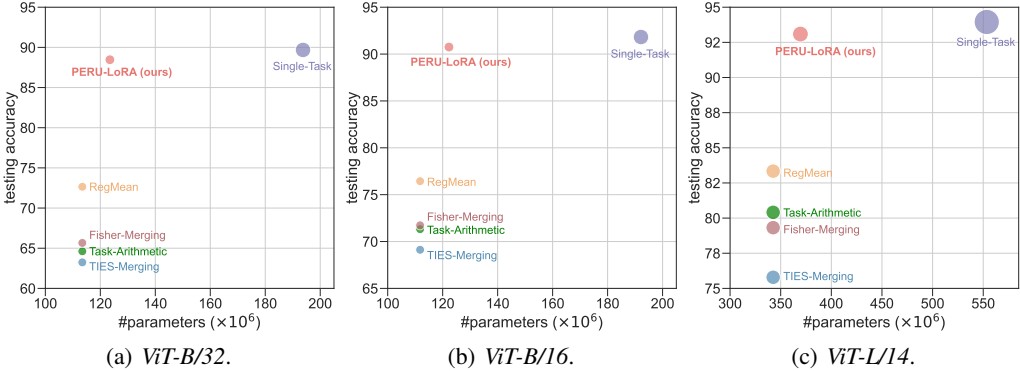

(a) *ViT-B/32*.     (b) *ViT-B/16*.     (c) *ViT-L/14*.

Figure 2: Testing accuracy (averaged over eight tasks) by reusing LoRA fine-tuned models (PERU-LoRA with $q = 16$).

We propose to approximate $\mathbf{A}_t \mathbf{B}_t^\top$ by a lower-rank matrix to save more parameters. Specifically, we first perform SVD for $\mathbf{A}_t \mathbf{B}_t^\top = \mathbf{U}_t \boldsymbol{\Sigma}_t \mathbf{V}_t^\top$, where $\mathbf{U}_t \in \mathbb{R}^{d_{\text{out}} \times r}$, $\mathbf{V}_t \in \mathbb{R}^{d_{\text{in}} \times r}$, and $\boldsymbol{\Sigma}_t \in \mathbb{R}^{r \times r}$ is a diagonal matrix with diagonal entries sorted from high to low. Let $\mathbf{U}_t(q) \in \mathbb{R}^{d_{\text{out}} \times q}$ be the submatrix of first $q$ columns of $\mathbf{U}_t$, $\mathbf{V}_t(q) \in \mathbb{R}^{d_{\text{in}} \times q}$ be the submatrix of first $q$ columns of $\mathbf{V}_t$, $\boldsymbol{\Sigma}_t(q) \in \mathbb{R}^{q \times q}$ be the submatrix of first $q$ rows and columns of $\boldsymbol{\Sigma}_t$ (corresponding to the $q$ largest singular values). The LoRA matrix $\mathbf{A}_t \mathbf{B}_t^\top$ is then approximated as $\mathbf{U}_t(q)\boldsymbol{\Sigma}_t(q)\mathbf{V}_t(q)^\top$, where the number of parameters is reduced from $r \times (d_{\text{out}} + d_{\text{in}})$ to $q \times (d_{\text{out}} + d_{\text{in}} + 1)$. $q$ can be much smaller than $r$, e.g., $q = 16$ compared with $r = 128$, saving $8\times$ additional parameters in LoRA matrices. In inference, $\boldsymbol{\theta}_0 + \mathbf{U}_t(q)\boldsymbol{\Sigma}_t(q)\mathbf{V}_t(q)^\top$ is used as the task model. The procedure, called PERU-LoRA, is shown in Algorithm 2.

---

[1]Experimental setup is in Section 4.1.

**Discussion.** Unlike reusing fully fine-tuned models, merging models before extracting a task-specific lower-rank matrix is infeasible to reuse LoRA fine-tuned models. Specifically, let $\boldsymbol{\theta}^\star$ be a merged model, then $\boldsymbol{\theta}_t - \boldsymbol{\theta}^\star = \boldsymbol{\theta}_0 + \mathbf{A}_t\mathbf{B}_t^\top - \boldsymbol{\theta}^\star$ is not always a rank-$r$ matrix. For example, when using Task-Arithmetic (Ilharco et al., 2023) as a merging algorithm, $\boldsymbol{\theta}_t - \boldsymbol{\theta}^\star = \sum_{t=1}^T \mathbf{A}_t\mathbf{B}^\top$, whose rank can be $qT$.

---

**Algorithm 2** PERU-LoRA.

---

**Require:** $\boldsymbol{\theta}_0$; LoRA matrices $\{(\mathbf{A}_t, \mathbf{B}_t)\}_{t=1}^T$; rank $q$;
1: **for** $t = 1, \ldots, T$ **do**
2:     compute $\mathbf{U}_t(q), \mathbf{V}_t(q), \boldsymbol{\Sigma}_t(q)$ from $\mathbf{A}_t\mathbf{B}_t^\top$ by SVD;
3:     evaluate $\boldsymbol{\theta}_0 + \mathbf{U}_t(q)\boldsymbol{\Sigma}_t(q)\mathbf{V}_t(q)^\top$ on task $t$'s testing set;
4: **end for**

---

# 4 EXPERIMENTS

## 4.1 EXPERIMENTS ON COMPUTER VISION TASKS

**Datasets and models.** Experiments are conducted on eight image classification tasks: *MNIST* (denoted MNI) (LeCun et al., 2010), *GTSRB* (denoted GTS) (Stallkamp et al., 2011), *SVHN* (denoted SVH) (Netzer et al., 2011), *RESISC45* (denoted RES) (Cheng et al., 2017), *SUN397* (denoted SUN) (Xiao et al., 2016), *EuroSAT* (denoted EUR) (Helber et al., 2019), *DTD* (Cimpoi et al., 2014), and *Cars* (denoted CAR) (Krause et al., 2013). Following Ilharco et al. (2023), we adopt three variants of the CLIP model (Radford et al., 2021) with *ViT* models (Dosovitskiy et al., 2021) including *ViT-B/32*, *ViT-B/16*, and *ViT-L/14* as image encoders. For PERU-FFT, we use the Task-Arithmetic (Ilharco et al., 2023) as the merging algorithm $\mathcal{A}_{\text{merging}}$.

**Baselines.** We compare with (i) Pre-Trained Model $\boldsymbol{\theta}_0$; (ii) Single-Task fully fine-tuned models (Single-Task); (iii) Multi-Task Learning (MTL) (Zhang & Yang, 2021) which requires all task data

Table 1: Testing accuracy on eight tasks reusing fully/LoRA fine-tuned models using *ViT-B/32*.

| | #params (M) | *MNI* | *GTS* | *SVH* | *RES* | *SUN* | *EUR* | *DTD* | *CAR* | *Avg* |
|---|---|---|---|---|---|---|---|---|---|---|
| Pre-Trained | 113 | 48.25 | 32.56 | 31.61 | 60.65 | 63.18 | 45.11 | 43.99 | 59.74 | 48.14 |
| Single-Task | 908 | 99.72 | 99.23 | 97.42 | 95.56 | 75.03 | 99.00 | 79.47 | 78.73 | 90.52 |
| MTL | 113 | 99.45 | 98.91 | 95.80 | 93.90 | 72.85 | 98.22 | 77.87 | 74.44 | 88.93 |
| **Fully FT** | | | | | | | | | | |
| Task-Arithmetic | 113 | 93.27 | 65.99 | 71.62 | 71.57 | 63.63 | 78.41 | 51.76 | 61.50 | 69.72 |
| Fisher-Merging | 113 | 80.71 | 75.15 | 74.08 | 70.24 | 65.25 | 81.48 | 49.84 | 62.90 | 69.96 |
| RegMean | 113 | 92.55 | 65.12 | 75.48 | 75.56 | 65.72 | 84.33 | 56.01 | 64.54 | 72.41 |
| TIES-Merging | 113 | 97.79 | 75.30 | 84.10 | 70.71 | 59.24 | 75.89 | 53.51 | 58.72 | 71.91 |
| Post-Pruning (1%) | 123 | 58.41 | 40.61 | 39.38 | 67.08 | 66.63 | 56.26 | 48.83 | 63.95 | 55.14 |
| Post-Pruning (5%) | 159 | 95.82 | 78.61 | 74.35 | 83.67 | 71.60 | 85.81 | 62.39 | 72.73 | 78.12 |
| Post-Pruning (10%) | 204 | 99.17 | 95.30 | 93.85 | 92.13 | **74.39** | 96.37 | 71.97 | **77.09** | 87.53 |
| PERU-FFT (1%) | 123 | 96.17 | 76.33 | 79.27 | 78.03 | 66.88 | 84.89 | 58.03 | 65.99 | 75.70 |
| PERU-FFT (5%) | 159 | 99.12 | 92.66 | 91.86 | 88.48 | 71.35 | 94.85 | 67.77 | 73.08 | 84.90 |
| PERU-FFT (10%) | 204 | **99.49** | 97.57 | 95.92 | 93.00 | 73.52 | **97.63** | 72.98 | 76.92 | **88.38** |
| **LoRA FT** | | | | | | | | | | |
| Single-Task | 194 | 99.61 | 98.71 | 97.34 | 95.57 | 73.42 | 98.63 | 76.91 | 77.25 | 89.68 |
| Task-Arithmetic | 113 | 86.90 | 51.44 | 66.50 | 68.16 | 62.32 | 76.19 | 48.62 | 56.85 | 64.62 |
| Fisher-Merging | 113 | 86.71 | 53.85 | 62.44 | 71.19 | 65.16 | 72.67 | 50.37 | 62.88 | 65.66 |
| RegMean | 113 | 94.45 | 60.10 | 81.11 | 74.57 | 65.10 | 88.15 | 53.72 | 63.97 | 72.65 |
| TIES-Merging | 113 | 82.48 | 45.89 | 58.95 | 70.67 | 65.20 | 71.11 | 49.15 | 62.44 | 63.24 |
| PERU-LoRA (4) | 116 | 99.16 | 92.04 | 93.98 | 86.48 | 68.61 | 95.37 | 65.37 | 62.74 | 82.97 |
| PERU-LoRA (8) | 118 | 99.54 | 96.23 | 96.45 | 92.16 | 70.33 | 98.26 | 72.55 | 67.35 | 86.61 |
| PERU-LoRA (16) | 123 | **99.62** | 97.99 | 97.08 | 94.56 | 72.29 | 98.37 | 76.44 | **71.31** | 88.46 |

for training a model; and the state-of-the-art merging methods include    (iv) Task-Arithmetic (Ilharco et al., 2023) merges model parameters by uniform averaging;    (v) Fisher-Merging (Matena & Raffel, 2022) takes weighted averaging based on Fisher information matrix computed on the validation loss; (vi) RegMean (Jin et al., 2023) merges linear layers by solving a local linear regression problem on the validation data;    (vii) TIES-Merging (Yadav et al., 2023) trims the task vectors and resolves the sign disagreements before aggregating parameters.

**Results.** Tables 1, 2, and 3 shows the testing accuracy on eight data sets using *ViT-B/32*, *ViT-B/16*, and *ViT-L/14*, respectively. As can be seen, for reusing fully fine-tuned models, by keeping top-10% values, both PERU-FFT and Post-Pruning achieve comparable performance with Single-Task, but are more parameter-efficient ($4.5\times$ fewer parameters). PERU-FFT (with addition 1% parameters per task) consistently performs better than the existing merging models method by a large margin, demonstrating the effectiveness of injecting sparse task-specific vectors into the shared model. Compared with Post-Pruning, PERU-FFT achieves higher accuracy (averaged over eight tasks), showing that merging the task-specific models before pruning the task vectors is effective. PERU-FFT, which keeps top-1% values of task vectors, performs largely better than existing merging models.

Table 2: Testing accuracy on eight tasks reusing fully/LoRA fine-tuned models using *ViT-B/16*.

| | | #params (M) | *MNI* | *GTS* | *SVH* | *RES* | *SUN* | *EUR* | *DTD* | *CAR* | *Avg* |
|---|---|---|---|---|---|---|---|---|---|---|---|
| | Pre-Trained | 112 | 51.79 | 43.34 | 51.98 | 65.76 | 65.50 | 55.22 | 45.11 | 64.57 | 55.41 |
| **Fully FT** | Single-Task | 894 | 99.72 | 99.15 | 97.86 | 96.57 | 78.71 | 99.33 | 82.29 | 87.20 | 92.60 |
| | Task-Arithmetic | 112 | 97.35 | 71.39 | 80.50 | 75.71 | 67.88 | 82.63 | 52.34 | 70.74 | 74.82 |
| | Fisher-Merging | 112 | 94.52 | 61.21 | 73.24 | 75.25 | 68.54 | 80.41 | 50.74 | 69.94 | 71.73 |
| | RegMean | 112 | 96.93 | 70.26 | 83.79 | 77.60 | 69.10 | 88.85 | 54.63 | 71.67 | 76.60 |
| | TIES-Merging | 112 | 98.75 | 74.43 | 88.84 | 78.48 | 66.21 | 85.93 | 57.13 | 73.15 | 77.86 |
| | Post-Pruning (1%) | 121 | 60.94 | 47.66 | 60.54 | 73.97 | 68.52 | 66.15 | 49.63 | 69.29 | 62.09 |
| | Post-Pruning (5%) | 157 | 96.06 | 77.36 | 82.08 | 88.70 | 74.42 | 94.22 | 64.89 | 79.28 | 82.13 |
| | Post-Pruning (10%) | 201 | 99.32 | 94.83 | 94.43 | 94.62 | 77.00 | 98.44 | 76.01 | 84.62 | 89.91 |
| | PERU-FFT (1%) | 121 | 98.32 | 79.85 | 85.12 | 82.89 | 71.22 | 89.30 | 59.79 | 75.33 | 80.23 |
| | PERU-FFT (5%) | 157 | 99.38 | 92.91 | 93.90 | 92.60 | 74.99 | 97.11 | 71.12 | 81.72 | 87.97 |
| | PERU-FFT (10%) | 201 | **99.56** | **97.34** | **96.91** | **95.30** | **77.11** | **98.67** | **77.77** | **85.04** | **90.96** |
| **LoRA FT** | Single-Task | 192 | 99.77 | 99.11 | 97.72 | 96.21 | 76.63 | 98.89 | 79.95 | 86.27 | 91.82 |
| | Task-Arithmetic | 112 | 95.59 | 63.06 | 77.30 | 72.92 | 66.05 | 82.67 | 49.04 | 64.46 | 71.38 |
| | Fisher-Merging | 112 | 94.51 | 61.19 | 73.22 | 75.24 | 68.57 | 80.41 | 50.74 | 69.93 | 71.73 |
| | RegMean | 112 | 97.89 | 68.73 | 85.26 | 76.30 | 68.17 | 91.96 | 52.66 | 70.54 | 76.44 |
| | TIES-Merging | 112 | 90.69 | 54.52 | 71.18 | 74.41 | 68.02 | 77.59 | 48.56 | 67.98 | 69.12 |
| | PERU-LoRA (4) | 114 | 99.35 | 93.96 | 95.52 | 88.65 | 72.21 | 96.81 | 69.73 | 71.05 | 85.91 |
| | PERU-LoRA (8) | 117 | 99.64 | 97.51 | 97.16 | 93.40 | 73.55 | 98.52 | 76.12 | 76.72 | 89.08 |
| | PERU-LoRA (16) | 122 | **99.66** | **98.54** | **97.61** | **95.25** | **75.54** | **98.78** | **78.72** | **81.88** | **90.75** |

As for reusing LoRA fine-tuned models, we can see that PERU-LoRA (16) achieves comparable performance with Single-Task, but is more parameter-efficient ($1.6\times$ fewer parameters). Furthermore, compared with existing merging models methods, both PERU-LoRA (8) and PERU-LoRA (16) by a large margin, while PERU-LoRA (2) also has a higher accuracy, demonstrating that extracting a lower task-specific matrix from the LoRA matrix is effective. Compared with PERU-FFT (10%) and Post-Pruning (10%), PERU-LoRA (16) performs better but has $1.7\times$ fewer parameters. Moreover, PERU-LoRA (16) achieves comparable performance with Single-Task (Fully FT) but has $7.4\times$ fewer parameters, showing that reusing the LoRA fine-tuned models is very effective and parameter-efficient. Furthermore, compared with the Pre-Trained model, PERU-LoRA (16) uses only 10M more parameters but almost double the accuracy for *ViT-B/32* and *ViT-B/16* models. As for *ViT-L/14*, PERU-LoRA (16) uses only 26M parameters but achieves $1.4\times$ higher accuracy than the Pre-Trained model.

Table 3: Testing accuracy on eight tasks reusing fully/LoRA fine-tuned models using *ViT-L/14*.

| | | #params (M) | *MNI* | *GTS* | *SVH* | *RES* | *SUN* | *EUR* | *DTD* | *CAR* | Avg |
|---|---|---|---|---|---|---|---|---|---|---|---|
| | Pre-Trained | 343 | 76.36 | 50.55 | 58.45 | 71.05 | 68.28 | 62.41 | 55.32 | 77.73 | 65.02 |
| Fully FT | Single-Task | 2,740 | 99.77 | 99.33 | 98.12 | 97.30 | 82.13 | 99.26 | 84.68 | 92.36 | 94.12 |
| | MTL | 343 | 99.63 | 99.07 | 97.57 | 96.32 | 80.84 | 99.19 | 84.36 | 90.64 | 93.45 |
| | Task-Arithmetic | 343 | 98.95 | 85.80 | 87.20 | 86.60 | 73.84 | 94.48 | 65.69 | 83.68 | 84.53 |
| | Fisher-Merging | 343 | 96.98 | 69.43 | 78.20 | 82.33 | 72.18 | 91.04 | 62.07 | 82.43 | 79.33 |
| | RegMean | 343 | 98.42 | 81.37 | 88.03 | 85.27 | 72.77 | 95.37 | 65.74 | 84.09 | 83.88 |
| | TIES-Merging | 343 | 99.01 | 81.34 | 89.42 | 89.49 | 76.18 | 95.96 | 68.24 | 86.83 | 85.81 |
| | Post-Pruning (1%) | 370 | 88.11 | 57.55 | 67.26 | 78.27 | 71.40 | 75.78 | 59.89 | 82.04 | 72.54 |
| | Post-Pruning (5%) | 480 | 99.07 | 84.66 | 87.85 | 92.75 | 77.40 | 97.48 | 72.02 | 88.96 | 87.52 |
| | Post-Pruning (10%) | 617 | 99.67 | 96.95 | 96.86 | 96.25 | 80.56 | **99.04** | 79.31 | **91.54** | 92.52 |
| | PERU-FFT (1%) | 370 | 99.17 | 90.67 | 90.99 | 89.62 | 75.55 | 96.30 | 69.36 | 86.06 | 87.21 |
| | PERU-FFT (5%) | 480 | 99.62 | 96.46 | 95.87 | 94.41 | 78.90 | 98.41 | 76.76 | 89.14 | 91.20 |
| | PERU-FFT (10%) | 617 | **99.74** | **98.43** | **97.43** | **96.37** | **80.79** | 98.93 | **80.53** | 90.72 | **92.87** |
| LoRA FT | Single-Task | 553 | 99.78 | 99.28 | 98.02 | 97.13 | 81.79 | 99.04 | 84.52 | 92.08 | 93.95 |
| | Task-Arithmetic | 343 | 97.59 | 72.35 | 81.47 | 83.03 | 72.40 | 91.59 | 62.45 | 82.42 | 80.41 |
| | Fisher-Merging | 343 | 96.98 | 69.40 | 78.18 | 82.32 | 72.18 | 91.00 | 62.07 | 82.43 | 79.32 |
| | RegMean | 343 | 98.53 | 80.39 | 84.83 | 85.70 | 72.90 | 95.41 | 65.05 | 83.93 | 83.34 |
| | TIES-Merging | 343 | 94.72 | 61.36 | 74.20 | 79.43 | 71.22 | 84.00 | 60.05 | 81.36 | 75.79 |
| | PERU-LoRA (4) | 349 | 99.53 | 97.47 | 96.98 | 93.32 | 76.61 | 98.63 | 76.33 | 84.07 | 90.37 |
| | PERU-LoRA (8) | 356 | 99.76 | 98.48 | 97.80 | 95.75 | 78.23 | 98.81 | 80.85 | 87.53 | 92.15 |
| | PERU-LoRA (16) | 369 | **99.78** | **98.92** | **98.02** | **96.56** | **79.91** | **99.04** | **82.93** | **89.55** | **93.09** |

(a) TaskArith.   (b) FisherMerg.   (c) RegMean.   (d) TiesMerg.   (e) Post-Pruning.   (f) PERU-FFT.

Figure 3: t-SNE of samples from *EuroSAT* for methods reusing fully fine-tuned *ViT-B/32* Models.

Figure 3 visualize the t-SNE (Van der Maaten & Hinton, 2008) of embeddings extracted from 200 images (20 images per class) randomly sampled from *EuroSAT* for methods reusing fully fine-tuned *ViT-B/32* models. As can be seen, both PERU-FFT (10%) and Post-Pruning (10%) have more compact and separable structures than existing merging models methods, demonstrating that injecting sparse task-specific vectors into the shared model is effective in extracting more discriminative features. Furthermore, clusters of PERU-FFT are denser than Post-Pruning.

Figure 4 visualize the t-SNE of embeddings extracted from 200 images (20 images per class) randomly sampled from *EuroSAT* for methods reusing LoRA fine-tuned *ViT-B/32* models. As can be seen, PERU-LoRA (16) has a more compact and separable structure than existing merging models methods, showing that using a lower rank to approximate the trained LoRA matrix (whose rank is 128) is effective in extracting discriminative features for classification.

## 4.2 EXPERIMENTS ON NATURAL LANGUAGE PROCESS TASKS

We conduct experiments on four standard text classification data sets: *MRPC* (Dolan et al., 2004), *RTE* (Wang et al., 2018), *SST-2* (Socher et al., 2013), and *QNLI* (Wang et al., 2018). We adopt *Flan-T5-base* (Chung et al., 2022) as the model for text classification.

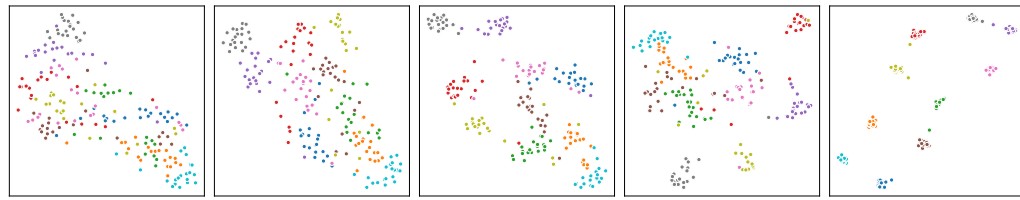

(a) TaskArithmetic.   (b) FisherMerging.   (c) RegMean.   (d) TiesMerging.   (e) PERU-LoRA.

Figure 4: t-SNE of samples from *EuroSAT* for methods reusing LoRA fine-tuned *ViT-B/32* Models.

Table 4 shows the testing accuracy. As can be seen, for reusing fully fine-tuned models, by keeping top-10% values, both PERU-FFT and Post-Pruning achieve comparable performance with Single-Task, but are much more parameter-efficient ($2.8\times$ few parameters). Furthermore, PERU-FFT outperforms Task-Arithmetic, showing that introducing sparse task-specific vectors to the merged model is better. Compared with Post-Pruning, PERU-FFT is better, demonstrating that merging models is effective in extracting shared knowledge before pruning task vectors. In particular, PERU-FFT with top-5% is better than Post-Pruning with top-10%. Hence, performing merging models is useful before extracting sparse task-specific vectors.

As for reusing LoRA fine-tuned models, PERU-LoRA with $q = 8$ or 16 achieves almost the same performance as Single-Task (LoRA FT) but has fewer parameters. Furthermore, PERU-LoRA outperforms existing merging methods by a large margin. Moreover, the performance of PERU-LoRA with $q = 8$ is close to that of Single-Task (Full FT) but is much more parameter-efficient ($3.9\times$ fewer parameters).

Table 4: Testing accuracy on four tasks reusing fully/LoRA fine-tuned models using *Flan-T5-base*.

| | | #params (M) | MRPC | RTE | SST-2 | QNLI | Avg |
|---|---|---|---|---|---|---|---|
| | Pre-Trained | 225 | 75.33 | 57.04 | 52.64 | 66.59 | 62.90 |
| **Fully FT** | Single-Task | 894 | 89.30 | 79.06 | 94.72 | 93.00 | 89.02 |
| | Task-Arithmetic | 225 | 82.29 | 73.29 | 93.23 | 88.16 | 84.24 |
| | Fisher-Merging | 225 | 80.61 | 70.04 | 92.66 | 85.63 | 82.23 |
| | RegMean | 225 | 84.52 | 76.53 | 92.55 | 91.16 | 86.19 |
| | TIES-Merging | 225 | 86.70 | 74.73 | 93.23 | 84.13 | 84.70 |
| | Post-Pruning (1%) | 234 | 75.52 | 62.45 | 69.72 | 81.90 | 72.40 |
| | Post-Pruning (5%) | 270 | 81.23 | 68.23 | 92.66 | 90.28 | 83.10 |
| | Post-Pruning (10%) | 314 | 86.26 | 77.62 | 94.04 | 91.69 | 87.40 |
| | PERU-FFT (1%) | 234 | 83.62 | 75.81 | 93.81 | 89.86 | 85.77 |
| | PERU-FFT (5%) | 270 | 86.63 | 78.34 | 94.04 | 91.43 | 87.61 |
| | PERU-FFT (10%) | 314 | **87.58** | **78.70** | **94.27** | **91.84** | **88.10** |
| **LoRA FT** | Single-Task | 239 | 87.47 | 79.06 | 94.04 | 92.70 | 88.32 |
| | Task-Arithmetic | 225 | 81.52 | 72.92 | 92.43 | 86.78 | 83.42 |
| | Fisher-Merging | 225 | 80.92 | 72.92 | 92.09 | 85.28 | 82.80 |
| | RegMean | 225 | 82.00 | 75.09 | 92.20 | 90.68 | 84.99 |
| | TIES-Merging | 225 | 83.47 | 65.34 | 92.32 | 82.92 | 81.01 |
| | PERU-LoRA (4) | 227 | 87.24 | 77.26 | 93.81 | 92.51 | 87.70 |
| | PERU-LoRA (8) | 229 | **87.64** | 78.70 | 93.92 | 92.53 | 88.20 |
| | PERU-LoRA (16) | 232 | 86.82 | **79.42** | **94.04** | **92.55** | **88.21** |

### 4.3 USEFULNESS OF INTEGRATING PERU-FFT INTO EXISTING MERGING METHODS

The proposed PERU-FFT is general and can be combined with any existing merging models methods. In Section 4.1, we use Task-Arithmetic as $\mathcal{A}_{\text{merging}}$ in Algorithm 1. We conduct additional experiments using the setting with ViT-B/32 to verify the benefits of integrating PERU-FFT into any other merging models methods. Figure 5 shows the testing accuracy (detailed results are shown in Table 5 of Appendix B.1). As can be seen, PERU-FFT consistently boosts the performance of existing methods by a large margin (Task-Arithmetic, Fisher-Merging, RegMean, TIES-Merging).

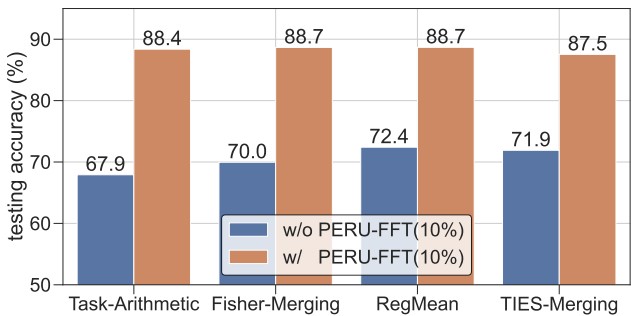 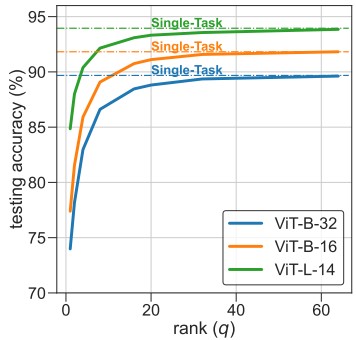

Figure 5: Effects of integrating PERU-FFT into existing merging models method.

Figure 6: Curves of average accuracy w.r.t. rank (q) in PERU-LoRA.

## 4.4 EFFECTS OF $q$ ON PERU-LORA

We perform experiments to study the effects of rank $q$ on the testing accuracy of PERU-LoRA using the settings in Section 4.1. Figure 6 shows the testing accuracy (averaged over eight tasks) w.r.t. $q$. As can be seen, increasing $q$ leads to a better performance. Furthermore, PERU-LoRA with rank-40 achieves almost the same performance as Single-Task (LoRA Fine-Tuned). Hence, using a lower-rank matrix to approximate the trained LoRA matrix is effective and more parameter-efficient.

## 4.5 EFFECTS OF $m\%$ ON POST-PRUNING AND PERU-FFT

In this section, we conduct experiments to study the effects of $m\%$ on the performance of Post-Pruning and PERU-FFT using the settings in Section 4.1. Figure 7 shows the testing accuracy (averaged over eight tasks) w.r.t. $m\% \in [0\%, 40\%]$ using *ViT-B/32*, *ViT-B/16*, and *ViT-L/14*. As can be seen, the accuracy of Post-Pruning and PERU-FFT increase when $m\%$ increases. When $m\%$ is larger than 20%, their accuracies reach the Single-Task performance and saturates. As for $m\% \leq 10\%$, PERU-FFT always performs better than Post-Pruning, suggesting that merging models before pruning is important when pruning most parameters.

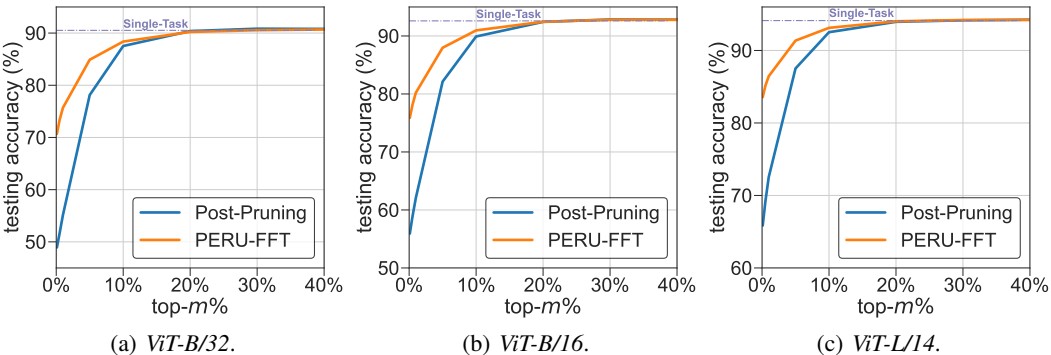

(a) *ViT-B/32*.          (b) *ViT-B/16*.          (c) *ViT-L/14*.

Figure 7: Curves of accuracy (averaged over eight tasks) w.r.t. top-$m\%$ values of task vectors.

## 5 CONCLUSION

In this paper, we studied the problem of reusing fine-tuned models. We proposed two parameter-efficient methods: (i) PERU-FFT for reusing fully fine-tuned models by injecting sparse task-specific vectors into the merged model; and (ii) PERU-LoRA for reusing LoRA fine-tuned models by using a lower rank matrix to approximate the LoRA matrix. Extensive experiments on computer vision and natural language processing tasks demonstrate that PERU-FFT and PERU-LoRA outperform existing merging methods significantly. Additionally, the proposed methods achieve comparable performance to Single-Task fine-tuned models but are much more parameter-efficient. Moreover, PERU-FFT is general and can be combined with any existing merging algorithms to boost performance.

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

## A    LIMITATIONS AND FUTURE WORKS

Like existing merging models methods (Task-Arithmetic, Fisher-Merging, RegMean, TIES-Merging), the proposed PERU is only suitable for task-specific models with the same architecture. Another limitation is that PERU requires the task-specific models to be finetuned from the same pre-trained model, which is also necessary for existing merging models methods. Existing merging methods and PERU need to know the specification of dataset sources when addressing different classification tasks. Weakening these assumptions is an important research direction in merging models.

## B    ADDITIONAL EXPERIMENTS

### B.1    COMBING PERU-FFT WITH EXISTING MERGING MODELS METHODS

The proposed PERU-FFT is general and can be combined with any existing merging models methods. In Section 4.1, we combine PERU-FFT method with Task-Arithmetic. We conducted additional experiments using the setting with *ViT-B/32* to verify the compatibility of PERU-FFT with existing merging models methods. Table 5 shows the testing accuracy. As we can see, PERU-FFT is consistently beneficial to existing methods (Task-Arithmetic, Fisher-Merging, RegMean, TIES-Merging).

Table 5: Accuracy on eight tasks with *ViT-B/32* when combing the proposed PERU-FFT with existing merging models methods.

| Method | #params (M) | MNI | GTS | SVH | RES | SUN | EUR | DTD | CAR | Avg |
|---|---|---|---|---|---|---|---|---|---|---|
| Task-Arithmetic | 113 | 93.27 | 65.99 | 71.62 | 71.57 | 63.63 | 78.41 | 51.76 | 61.50 | 69.72 |
| Task-Arithmetic + PERU-FFT (1%) | 123 | 96.17 | 76.33 | 79.27 | 78.03 | 66.88 | 84.89 | 58.03 | 65.99 | 75.70 |
| Task-Arithmetic + PERU-FFT (5%) | 159 | 99.12 | 92.66 | 91.86 | 88.48 | 71.35 | 94.85 | 67.77 | 73.08 | 84.90 |
| Task-Arithmetic + PERU-FFT (10%) | 204 | **99.49** | **97.57** | **95.92** | **93.00** | **73.52** | **97.63** | **72.98** | **76.92** | **88.38** |
| Fisher-Merging | 113 | 80.71 | 75.15 | 74.08 | 70.24 | 65.25 | 81.48 | 49.84 | 62.90 | 69.96 |
| Fisher-Merging + PERU-FFT (1%) | 123 | 92.29 | 69.23 | 71.50 | 77.54 | 68.27 | 79.59 | 56.28 | 67.40 | 72.76 |
| Fisher-Merging + PERU-FFT (5%) | 159 | 98.81 | 91.39 | 90.31 | 88.73 | 72.38 | 94.33 | 67.18 | 74.26 | 84.67 |
| Fisher-Merging + PERU-FFT (10%) | 204 | **99.46** | **97.26** | **95.73** | **93.46** | **74.43** | **97.37** | **73.72** | **77.81** | **88.66** |
| RegMean | 113 | 92.55 | 65.12 | 75.48 | 75.56 | 65.72 | 84.33 | 56.01 | 64.54 | 72.41 |
| RegMean + PERU-FFT (1%) | 123 | 93.79 | 71.46 | 78.77 | 77.95 | 67.47 | 87.15 | 58.14 | 66.47 | 75.15 |
| RegMean + PERU-FFT (5%) | 159 | 98.59 | 91.83 | 91.74 | 88.65 | 72.01 | 96.19 | 67.98 | 73.77 | 85.10 |
| RegMean + PERU-FFT (10%) | 204 | **99.37** | **97.39** | **95.76** | **93.56** | **74.37** | **97.89** | **74.04** | **77.09** | **88.68** |
| TIES-Merging | 113 | 97.79 | 75.30 | 84.10 | 70.71 | 59.24 | 75.89 | 53.51 | 58.72 | 71.91 |
| TIES-Merging + PERU-FFT (1%) | 123 | 98.82 | 86.25 | 87.27 | 75.79 | 61.29 | 87.15 | 58.78 | 62.22 | 77.20 |
| TIES-Merging + PERU-FFT (5%) | 159 | 99.44 | 96.42 | 94.52 | 86.86 | 67.01 | 95.22 | 67.45 | 70.41 | 84.67 |
| TIES-Merging + PERU-FFT (10%) | 204 | **99.65** | **98.22** | **96.40** | **91.86** | **70.32** | **97.22** | **72.45** | **74.12** | **87.53** |

## B.2 ABLATION STUDY ON COMPRESSING LORA MATRICES

We perform singular value decomposition to $\mathbf{A}_t\mathbf{B}_t^\top$ as it can obtain the best rank-$k$ approximation of $\mathbf{A}_t\mathbf{B}_t^\top$, i.e., $\arg\min_{\text{rank}(\mathbf{C})\leq k} \|\mathbf{C} - \mathbf{A}_t\mathbf{B}_t^\top\|_F$. Note that $\mathbf{A}_t\mathbf{B}_t^\top$ is applied to the pre-trained weight $\boldsymbol{\theta}_0$ (i.e., $\boldsymbol{\theta}_t = \boldsymbol{\theta}_0 + \mathbf{A}_t\mathbf{B}_t^\top$), thus, approximating $\mathbf{A}_t\mathbf{B}_t^\top$ might be more effective than approximating $\mathbf{A}_t$ and $\mathbf{B}_t$ separately. We conducted an ablation experiment with *ViT-B/32* using the setting in Section 4.1. Table below shows the testing accuracy of $\boldsymbol{\theta}_0 + \text{Approx}(\mathbf{A}_t\mathbf{B}_t^\top)$ with $\boldsymbol{\theta}_t = \boldsymbol{\theta}_0 + \text{Approx}(\mathbf{A}_t)\text{Approx}(\mathbf{B}_t)^\top$. As can be seen, $\text{Approx}(\mathbf{A}_t\mathbf{B}_t^\top)$ **consistently performs better** than $\text{Approx}(\mathbf{A}_t)\text{Approx}(\mathbf{B}_t)^\top$.

Table 6: Testing accuracy on eight tasks of $\text{Approx}(\mathbf{A}_t)\text{Approx}(\mathbf{B}_t)^\top$ and $\text{Approx}(\mathbf{A}_t\mathbf{B}_t^\top)$ when reusing LoRA fine-tuned models with *ViT-B/32*.

| | rank | MNI | GTS | SVH | RES | SUN | EUR | DTD | CAR | Avg |
|---|---|---|---|---|---|---|---|---|---|---|
| $\text{Approx}(\mathbf{A}_t)\text{Approx}(\mathbf{B}_t)^\top$ | 4 | 61.42 | 38.29 | 39.17 | 64.41 | 64.35 | 64.74 | 45.85 | 60.63 | 54.86 |
| $\text{Approx}(\mathbf{A}_t\mathbf{B}_t^\top)$ | 4 | **99.16** | **92.04** | **93.98** | **86.48** | **68.61** | **95.37** | **65.37** | **62.74** | **82.97** |
| $\text{Approx}(\mathbf{A}_t)\text{Approx}(\mathbf{B}_t)^\top$ | 8 | 79.39 | 45.81 | 50.33 | 70.73 | 65.50 | 79.67 | 48.35 | 62.09 | 62.73 |
| $\text{Approx}(\mathbf{A}_t\mathbf{B}_t^\top)$ | 8 | **99.54** | **96.23** | **96.45** | **92.16** | **70.33** | **98.26** | **72.55** | **67.35** | **86.61** |
| $\text{Approx}(\mathbf{A}_t)\text{Approx}(\mathbf{B}_t)^\top$ | 16 | 96.25 | 72.49 | 81.47 | 81.67 | 67.40 | 93.56 | 57.18 | 66.22 | 77.03 |
| $\text{Approx}(\mathbf{A}_t\mathbf{B}_t^\top)$ | 16 | **99.62** | **97.99** | **97.08** | **94.56** | **72.29** | **98.37** | **76.44** | **71.31** | **88.46** |

## B.3 ABLATION STUDY ON NUMBER OF TASKS

We conducted an ablation experiment on **four** computer vision tasks with *ViT-B/32*. Table 7 below shows the testing accuracy. As can be seen, for merging fully finetuned models, with a small number of parameters, Post-Pruning (5%) and Post-Pruning (10%) outperform existing merging methods (Task-Arithmetic, Fisher-Merging, RegMean, TIES-Merging), demonstrating the effectiveness of introducing sparse task-specific vectors into the shared model. Compared with Post-Pruning, PERU-FFT achieves higher accuracy (averaged over four tasks), suggesting that merging the task-specific models before pruning the task vectors is effective. Moreover, compared with Single-Task method, PERU-FFT (10%) is more parameter-efficient and achieves comparable performance. For merging LoRA finetuned models, PERU-LoRA achieves higher accuracy (averaged over four tasks) than previous merging methods. **These observations are consistent with results on eight computer vision tasks in Table 1.**

Table 7: Testing accuracy on four tasks reusing fully/LoRA fine-tuned models using *ViT-B/32*.

| | | #params (M) | SVH | EUR | DTD | CAR | Avg |
|---|---|---|---|---|---|---|---|
| | Pre-Trained | 113 | 31.61 | 45.11 | 43.99 | 59.74 | 45.11 |
| **Fully FT** | Single-Task | 452 | 97.42 | 99.00 | 79.47 | 78.73 | 88.66 |
| | Task-Arithmetic | 113 | 77.30 | 92.78 | 61.49 | 67.83 | 74.85 |
| | Fisher-Merging | 113 | 72.02 | 89.89 | 58.72 | 67.50 | 72.03 |
| | RegMean | 113 | 86.34 | 95.78 | 64.41 | 69.11 | 78.91 |
| | TIES-Merging | 113 | 88.95 | 94.11 | 66.70 | 70.68 | 80.11 |
| | Post-Pruning (1%) | 118 | 39.39 | 56.26 | 48.78 | 63.95 | 52.05 |
| | Post-Pruning (5%) | 136 | 74.35 | 85.81 | 62.39 | 72.74 | 73.83 |
| | Post-Pruning (10%) | 159 | 93.85 | 96.37 | 71.91 | 77.09 | 84.81 |
| | PERU-FFT (1%) | 118 | 84.48 | 95.04 | 64.95 | 70.87 | 78.84 |
| | PERU-FFT (5%) | 136 | 94.45 | 97.41 | 71.81 | 75.69 | 84.84 |
| | PERU-FFT (10%) | 159 | **96.68** | **98.15** | **75.53** | **77.89** | **87.06** |
| **LoRA FT** | Single-Task | 153 | 97.34 | 98.63 | 76.91 | 77.25 | 87.52 |
| | Task-Arithmetic | 113 | 62.16 | 80.78 | 53.35 | 64.08 | 65.09 |
| | Fisher-Merging | 113 | 71.94 | 89.81 | 58.72 | 67.54 | 72.01 |
| | RegMean | 113 | 89.33 | 94.85 | 61.60 | 67.57 | 78.34 |
| | TIES-Merging | 113 | 47.16 | 67.52 | 48.30 | 62.21 | 56.29 |
| | PERU-LoRA (4) | 114 | 93.98 | 95.37 | 65.37 | 62.74 | 79.37 |
| | PERU-LoRA (8) | 116 | 96.45 | 98.26 | 72.55 | 67.35 | 83.66 |
| | PERU-LoRA (16) | 118 | **97.08** | **98.37** | **76.44** | **71.31** | **85.80** |

## B.4 EXPERIMENTS ON *VTAB*

We conducted an experiment on the *Natural* group of *VTAB* (Zhai et al., 2019) using *ViT-B/16*, where tasks are more similar to each other. Table 8 shows the testing accuracy. The observations are **consistent** with the experimental results in Section 4.1. Specifically, as we can see, by keeping top-10% values, both PERU-FFT and Post-Pruning achieve comparable performance with Single-Task, but are more parameter-efficient (4.5× fewer parameters). PERU-FFT (with an additional 1% of parameters per task) performs better than the existing merging models method, showing the effectiveness of introducing sparse task-specific vectors into the merged model. Compared with Post-Pruning, PERU-FFT achieves higher accuracy (averaged over seven tasks), showing that merging the task-specific models before pruning the task vectors is more effective.

Table 8: Accuracy on seven tasks from the *Natural* group of *VTAB*.

| | #params (M) | CIF | CAL | DTD | FLO | PET | SVH | SUN | Avg |
|---|---|---|---|---|---|---|---|---|---|
| Pre-Trained | 112 | 66.91 | 82.76 | 45.11 | 71.33 | 87.19 | 51.98 | 65.50 | 67.25 |
| Single-Task | 894 | 90.23 | 97.20 | 82.29 | 94.88 | 94.55 | 97.86 | 78.71 | 90.82 |
| Task-Arithmetic | 112 | 83.33 | 87.07 | 52.87 | 66.87 | 89.10 | 83.52 | 67.01 | 75.68 |
| Fisher-Merging | 112 | 80.81 | 86.96 | 51.28 | 73.72 | 89.51 | 69.92 | 69.32 | 74.50 |
| RegMean | 112 | 81.32 | 87.07 | 55.53 | 75.41 | 90.38 | 84.94 | 69.88 | 77.79 |
| TIES-Merging | 112 | 82.77 | 87.69 | 57.39 | 70.39 | 89.59 | 88.28 | 67.42 | 77.65 |
| Post-Pruning (1%) | 121 | 74.37 | 85.06 | 49.63 | 73.59 | 88.23 | 60.53 | 68.51 | 71.42 |
| Post-Pruning (5%) | 157 | 86.55 | 90.60 | 64.89 | 78.45 | 91.58 | 82.06 | 74.41 | 81.22 |
| Post-Pruning (10%) | 201 | 89.61 | 93.68 | 76.01 | **84.34** | **94.41** | 94.42 | **77.00** | 87.07 |
| PERU-FFT (1%) | 121 | 85.39 | 89.93 | 58.30 | 70.89 | 91.61 | 87.78 | 69.76 | 79.09 |
| PERU-FFT (5%) | 157 | 88.50 | 92.56 | 69.73 | 77.62 | 93.59 | 94.71 | 74.18 | 84.41 |
| PERU-FFT (10%) | 201 | **89.75** | **93.90** | **76.22** | 83.98 | 94.06 | **97.01** | 76.46 | **87.34** |

## B.5 EXPERIMENTS USING CNN-BASED MODELS

We conducted an additional experiment using a CNN-based model *ConvNeXt-Base* (Liu et al., 2022) on eight computer vision tasks. Table 9 shows the testing accuracy. The following observations are

Table 9: Testing accuracy on eight tasks reusing fully fine-tuned models using *ConvNeXt-Base*.

| | #params (M) | MNI | GTS | SVH | RES | SUN | EUR | DTD | CAR | Avg |
|---|---|---|---|---|---|---|---|---|---|---|
| Pre-Trained | 179 | 64.39 | 46.56 | 53.73 | 65.94 | 71.61 | 52.37 | 61.54 | 91.24 | 63.42 |
| Single-Task | 1,435 | 99.78 | 99.22 | 98.01 | 96.67 | 80.49 | 99.19 | 85.74 | 94.91 | 94.25 |
| Task-Arithmetic | 179 | 97.73 | 81.31 | 82.96 | 76.56 | 72.12 | 78.07 | 68.40 | 92.87 | 81.25 |
| Fisher-Merging | 179 | 95.47 | 67.67 | 77.93 | 76.21 | 72.80 | 74.22 | 67.82 | 92.53 | 78.08 |
| RegMean | 179 | 97.92 | 81.25 | 86.47 | 80.65 | 74.00 | 89.26 | 72.55 | 93.82 | 84.49 |
| TIES-Merging | 179 | 99.16 | 85.32 | 88.83 | 72.97 | 69.44 | 78.37 | 65.27 | 91.93 | 81.41 |
| Post-Pruning (1%) | 194 | 85.64 | 54.60 | 65.75 | 73.29 | 73.68 | 65.30 | 67.13 | 92.54 | 72.24 |
| Post-Pruning (5%) | 251 | 99.04 | 83.80 | 89.97 | 87.43 | 77.26 | 91.11 | 76.33 | 94.28 | 87.40 |
| Post-Pruning (10%) | 323 | 99.65 | 95.65 | 96.47 | 93.21 | **79.29** | **97.89** | 80.74 | 94.86 | 92.22 |
| PERU-FFT (1%) | 194 | 99.02 | 88.49 | 87.86 | 81.89 | 73.79 | 86.26 | 72.07 | 93.56 | 85.37 |
| PERU-FFT (5%) | 251 | 99.60 | 95.90 | 95.01 | 90.08 | 76.57 | 95.00 | 78.09 | 94.68 | 90.62 |
| PERU-FFT (10%) | 323 | **99.72** | **98.02** | **97.33** | **93.65** | 78.66 | 97.41 | **82.39** | **95.04** | **92.78** |

**consistent** with results obtained from ViT-based models in Section 4.1. As we can see, (i) compared with Single-Task method, both PERU-FFT (10%) and Post-Pruning (10%) achieve comparable performance but have $4.5\times$ fewer parameters. With an additional 1% of parameters per task, PERU-FFT performs better than the existing merging models method, showing that introducing sparse task-specific vectors into the merged model is effective. PERU-FFT consistently achieves higher accuracy (averaged over seven tasks) than Post-Pruning, showing that merging the task-specific models before pruning the task vectors is more effective.

## B.6 ABLATION STUDY ON PRUNING METHODS

We conducted additional experiments with *ViT-B/32* to compare the performance of pruning $\theta_t$ and pruning $\mathbf{v}_t$ or $\mathbf{u}_t$. Table 10 shows the testing accuracy, where Pruning $\theta_t$ (m%) denotes keeping the top-m% parameters in $\theta_t$. As can be seen, pruning $\theta_t$ is not effective. For example, Pruning $\theta_t$ (50%) has very low accuracy. In contrast, keeping top-10% of $\mathbf{v}_t$ or $\mathbf{u}_t$ perform much better (+80%). Compared with Pruning $\theta_t$ (90%), PERU-FFT (10%) achieves comparable performance but has $4\times$ fewer parameters. Hence, pruning $\mathbf{u}_t$ is more effective and promising than pruning $\theta_t$.

Table 10: Comparison between pruning $\theta_t$ and pruning $\mathbf{v}_t$ or $\mathbf{u}_t$.

| | #params (M) | MNI | GTS | SVH | RES | SUN | EUR | DTD | CAR | Avg |
|---|---|---|---|---|---|---|---|---|---|---|
| Pre-Trained | 113 | 48.25 | 32.56 | 31.61 | 60.65 | 63.18 | 45.11 | 43.99 | 59.74 | 48.14 |
| Single-Task | 908 | 99.72 | 99.23 | 97.42 | 95.56 | 75.03 | 99.00 | 79.47 | 78.73 | 90.52 |
| Pruning $\theta_t$ (10%) | 91 | 9.82 | 0.70 | 8.49 | 3.00 | 0.25 | 9.52 | 1.60 | 0.60 | 4.25 |
| Pruning $\theta_t$ (20%) | 181 | 10.28 | 1.77 | 6.71 | 2.11 | 0.28 | 16.11 | 2.45 | 0.46 | 5.02 |
| Pruning $\theta_t$ (30%) | 271 | 9.91 | 3.72 | 17.62 | 2.63 | 0.44 | 10.78 | 2.23 | 0.49 | 5.98 |
| Pruning $\theta_t$ (40%) | 362 | 10.09 | 5.08 | 6.29 | 4.48 | 0.32 | 14.48 | 2.71 | 0.40 | 5.48 |
| Pruning $\theta_t$ (50%) | 452 | 10.94 | 5.59 | 20.45 | 6.73 | 0.92 | 17.00 | 7.23 | 0.53 | 8.67 |
| Pruning $\theta_t$ (60%) | 542 | 84.54 | 43.72 | 63.88 | 44.63 | 15.23 | 34.67 | 31.91 | 3.47 | 40.26 |
| Pruning $\theta_t$ (70%) | 632 | 98.83 | 80.37 | 91.32 | 77.48 | 48.49 | 70.11 | 56.22 | 38.75 | 70.20 |
| Pruning $\theta_t$ (80%) | 723 | 99.55 | 95.04 | 96.35 | 88.70 | 64.13 | 87.81 | 72.18 | 65.24 | 83.63 |
| Pruning $\theta_t$ (90%) | 814 | 99.69 | 99.06 | 97.39 | 95.24 | 73.59 | 98.81 | 79.10 | 77.08 | 89.99 |
| Post-Pruning $\mathbf{v}_t$ (1%) | 123 | 58.41 | 40.61 | 39.38 | 67.08 | 66.63 | 56.26 | 48.83 | 63.95 | 55.14 |
| Post-Pruning $\mathbf{v}_t$ (5%) | 159 | 95.82 | 78.61 | 74.35 | 83.67 | 71.60 | 85.81 | 62.39 | 72.73 | 78.12 |
| Post-Pruning $\mathbf{v}_t$ (10%) | 204 | 99.17 | 95.30 | 93.85 | 92.13 | 74.39 | 96.37 | 71.97 | 77.09 | 87.53 |
| PERU-FFT $\mathbf{u}_t$ (1%) | 123 | 96.17 | 76.33 | 79.27 | 78.03 | 66.88 | 84.89 | 58.03 | 65.99 | 75.70 |
| PERU-FFT $\mathbf{u}_t$ (5%) | 159 | 99.12 | 92.66 | 91.86 | 88.48 | 71.35 | 94.85 | 67.77 | 73.08 | 84.90 |
| PERU-FFT $\mathbf{u}_t$ (10%) | 204 | 99.49 | 97.57 | 95.92 | 93.00 | 73.52 | 97.63 | 72.98 | 76.92 | 88.38 |

