# OpenReview forum: "Effective and Parameter-Efficient Reusing Fine-Tuned Models"
_ICLR.cc/2024/Conference — ICLR 2024 Conference Withdrawn Submission_

### Official Review · Reviewer_a2T8 · 2023-10-28

**Soundness:** 3 good
**Presentation:** 3 good
**Contribution:** 2 fair
**Rating:** 6
**Confidence:** 4

**Summary:**

This paper addresses the storage challenges posed by models fine-tuned for multiple downstream tasks. It presents efficient methods for reusing fine-tuned models, specifically PERU-FFT which merges and prunes task-specific vectors of fully fine-tuned models, and PERU-LoRA which employs a lower rank matrix for approximating the LoRA matrix in LoRA fine-tuned models. Through experiments in computer vision and natural language processing, the proposed methods significantly surpass traditional model merging techniques in performance, and while being more parameter-efficient, they match the performance of dedicated task-specific fine-tuned models.

**Strengths:**

The paper is well written and easy to follow. The proposed methods are not only effective but are also parameter-efficient, making them valuable for real-world applications.

**Weaknesses:**

The primary limitation of the paper is its constrained novelty. The methodology seems to draw heavily from pre-existing techniques, and it lacks distinct contributions that set it apart. Please see questions for details.

**Questions:**

1.	The novelty of the proposed method appears somewhat constrained. At a glance, the approach seems to be an amalgamation of model merging, and parameter-efficient tuning and pruning. The good performance of the proposed method mainly comes from the task-specific parameters. It would be valuable for the authors to elaborate on any unique contributions or differentiating aspects of their method that go beyond this apparent synthesis.

2.	In Section 3.2, the authors have chosen to use singular value decomposition (SVD) to approximate the product $A_t B_t^{\top}$. An intuitive approach might be to directly reduce the rank of matrices $A_t$ and $B_t$. Can the authors provide insights into why this direct rank reduction was not pursued? Are there challenges or drawbacks associated with it that the proposed SVD method circumvents?

3.	In the proposed method, several points of clarification regarding the comparison with LoRA emerge. Firstly, it would be beneficial to understand what distinguishes the proposed method from LoRA. While the primary technique appears to focus on reducing the rank of $A_t B_t^{\top}$. Secondly, when referencing Table 4, one can observe that PERU-LoRA (16) has 232M parameters, which is only a marginal 3% reduction compared to the Single-Task model, yet its performance seems to lag behind. It raises the question of whether this slight reduction in parameters warrants the observed decrease in performance. Expounding on these aspects would provide a deeper understanding of the method's value proposition and potential areas for improvement.

4.	The proposed approach appears to have a broad compatibility with several existing model merging methods, including Fisher-Merging (Matena & Raffel, 2022), RegMean (Jin et al., 2023), and TIES-Merging (Yadav et al., 2023). It would be insightful to understand how the performance of the presented method fares when juxtaposed with these model merging methods.

5.	The supplementary material provided appears to be incorrect. It seems the authors mistakenly included the main text within the supplementary section.

---

> ### Author Response · Authors · 2023-11-21
> **Reply to Reviewer a2T8 (1/4)**
>
> Thanks for your thoughtful review and valuable feedback. We address your concerns as follows.
>
> > Q1. "The primary limitation of the paper is its constrained novelty. The methodology seems to draw heavily from pre-existing techniques, and it lacks distinct contributions that set it apart. The novelty of the proposed method appears somewhat constrained. At a glance, the approach seems to be an amalgamation of model merging, and parameter-efficient tuning and pruning. The good performance of the proposed method mainly comes from the task-specific parameters. It would be valuable for the authors to elaborate on any unique contributions or differentiating aspects of their method that go beyond this apparent synthesis."
>
> **A1.**
> (i) There is **a large gap** between the performance of existing merging models methods and the single-task method, as shown in Figure 1. Filling this gap is an important issue in merging models. We propose to **inject sparse task-specific vectors into the merged model**. The proposed method PERU-FFT achieves comparable performance to the single-task method but with higher parameter-efficiency. Moreover, PERU-FFT performs better than existing merging models methods. Our first attempt to fill this gap is a simple, effective, and reasonable method.
>
> (ii) For merging fully finetuned models, our PERU-FFT adds the top-20\% of $\theta_t - \theta^\star$ to the merged model $\theta^\star$, which **contains knowledge from other tasks**. Note that TIES-Merging adds the top-20% of $\theta_t-\theta_0$ (denoted $\hat{\bf v}_t$) to the pre-trained model $\theta_0$ (i.e., Post-Pruning in Algorithm 1 of the submission), thus, the pruned task model $\theta_0 + \hat{\bf v}_t$ **does not use any shared knowledge from other tasks**. As shown in Figure 1 of the submission, **PERU-FFT consistently performs better than Post-Pruning**, demonstrating that **sharing knowledge across tasks is more effective**.
>
> (ii) For merging LoRA-finetuned models, existing methods are not suitable as **they have not exploited the LoRA structure** (i.e., ${\bf A}_t{\bf B}_t^\top$). As shown in Figure 2 of the submission, existing methods perform much worse than the single-task method. Our proposed PERU-LoRA **uses singular value decomposition to compress the LoRA matrices** ${\bf A}_t{\bf B}_t^\top$. With the compressed LoRA matrices, PERU-LoRA performs much better than existing merging model methods, as shown in Figure 2. PERU-LoRA is a simple but effective method for merging LoRA-finetuned models.

---

> ### Author Response · Authors · 2023-11-21
> **Reply to Reviewer a2T8 (2/4)**
>
> > "Q2. In Section 3.2, the authors have chosen to use singular value decomposition (SVD) to approximate the product ${\bf A}_t{\bf B}_t^\top$. An intuitive approach might be to directly reduce the rank of matrices ${\bf A}_t$ and ${\bf B}_t$. Can the authors provide insights into why this direct rank reduction was not pursued? Are there challenges or drawbacks associated with it that the proposed SVD method circumvents?"
>
> **A2.** We perform singular value decomposition to ${\bf A}\_t{\bf B}\_t^\top$ as it can obtain the **best rank-$k$ approximation** of ${\bf A}\_t{\bf B}\_t^\top$, i.e., $\arg\min_{\text{rank}({\bf C})\leq k}\\| {\bf C} - {\bf A}\_t{\bf B}\_t^\top\\|_{\text{F}}^2$. Note that ${\bf A}\_t{\bf B}\_t^\top$ is applied to the pre-trained weight $\theta_0$ (i.e., $\theta_t=\theta_0+{\bf A}\_t{\bf B}\_t^\top$), thus, approximating ${\bf A}\_t{\bf B}\_t^\top$ might be more effective than approximating ${\bf A}\_t$ and ${\bf B}\_t$ separately. As suggested, we conducted an ablation experiment with *ViT-B/32* using the setting in Section 4.1. Table below compares the testing accuracy of $\theta_0+\text{Approx}({\bf A}\_t{\bf B}\_t^\top)$ with $\theta\_t=\theta_0 + \text{Approx}({\bf A}\_t)\text{Approx}({\bf B}\_t)^\top$. As can be seen, Approx(${\bf A}\_t{\bf B}\_t^\top$) **consistently performs better** than $\text{Approx}({\bf A}\_t)\text{Approx}({\bf B}\_t)^\top$. We added the results in Appendix B.2 of the updated paper.
>
> \begin{array}{lcccccccccc}
> \hline
>  & \text{rank} & \text{\\#params (M)} & \text{MNI} & \text{GTS} & \text{SVH} & \text{RES} & \text{SUN} & \text{EUR} & \text{DTD} & \text{CAR} & \text{Avg} \newline
> \hline
> \text{Approx}({\bf A}_t)\text{Approx}({\bf B}_t)^\top & 4 & 116 & 61.42 & 38.29 & 39.17 & 64.41 & 64.35 & 64.74 & 45.85 & 60.63 & 54.86 \newline
> \text{Approx(${\bf A}_t{\bf B}_t^\top$)} & 4 & 116 & {\bf 99.16} & {\bf 92.04} & {\bf 93.98} & {\bf 86.48} & {\bf 68.61} & {\bf 95.37} & {\bf 65.37} & {\bf 62.74} & {\bf 82.97} \newline
> \hline
> \text{Approx}({\bf A}_t)\text{Approx}({\bf B}_t)^\top & 8 & 118 & 79.39 & 45.81 & 50.33 & 70.73 & 65.50 & 79.67 & 48.35 & 62.09 & 62.73 \newline
> \text{Approx(${\bf A}_t{\bf B}_t^\top$)} & 8& 118 & {\bf 99.54} & {\bf 96.23} & {\bf 96.45} & {\bf 92.16} & {\bf 70.33} & {\bf 98.26} & {\bf 72.55} & {\bf 67.35} & {\bf 86.61} \newline
> \hline
> \text{Approx}({\bf A}_t)\text{Approx}({\bf B}_t)^\top & 16 & 123 & 96.25 & 72.49 & 81.47 & 81.67 & 67.40 & 93.56 & 57.18 & 66.22 & 77.03\newline
> \text{Approx(${\bf A}_t{\bf B}_t^\top$)} & 16 & 123 & {\bf 99.62} & {\bf 97.99} & {\bf 97.08} & {\bf 94.56} & {\bf 72.29} &  {\bf 98.37} &  {\bf 76.44} & {\bf 71.31} & {\bf 88.46} \newline
> \hline
> \end{array}
>
> ---
>
> > "Q3. In the proposed method, several points of clarification regarding the comparison with LoRA emerge. Firstly, it would be beneficial to understand what distinguishes the proposed method from LoRA. While the primary technique appears to focus on reducing the rank of ${\bf A}_t{\bf B}_t^\top$."
>
> **A3.**
> (i) LoRA is a method of **finetuning** pre-trained models on data, while PERU-LoRA is a method that aims at **merging** existing LoRA finetuned models into a multi-task model.
> (ii) LoRA learns the low rank matrices ${\bf A}_t$ and ${\bf B}_t$, while PERU-LoRA uses singular value decomposition to compress the learned ${\bf A}_t{\bf B}_t^\top$.
> (iii) **The goals of LoRA and PERU-LoRA are different**: the former aims to obtain task-specific models by training, and the latter aims to obtain a multi-task model by merging existing finetuned models without further training.
> (iv) As **LoRA** is for training, it **needs data**; however, the merging method **PERU-LoRA is training-free** and therefore does not require data.

---

> ### Author Response · Authors · 2023-11-21
> **Reply to Reviewer a2T8 (3/4)**
>
> > Q4. "when referencing Table 4, one can observe that PERU-LoRA (16) has 232M parameters, which is only a marginal 3\% reduction compared to the Single-Task model, yet its performance seems to lag behind. It raises the question of whether this slight reduction in parameters warrants the observed decrease in performance."
>
> **A4.** In Table 4, PERU-LoRA (16) achieves comparable performance to Single-Task (LoRA) but with 7M fewer parameters. In this setting, the reduction is not very significant as the Single-Task (LoRA) is already very parameter-efficient (as shown in Table 4, the number of parameters in Single-Task (LoRA) (239M) is close to that in the pre-trained model (225M)). Hence, **in this setting, there is only a small room for PERU-LoRA to improve the parameter-efficiency** of Single-Task (LoRA). Indeed, only 239M-225M=14M additional parameters can be reduced, and our PERU-LoRA (16) reduces half of them (239M-232M=7M). Note that **in other settings (Tables 1, 2, 3, or Figures 1, 2), compared with Single-Task (LoRA), PERU-LoRA (16) has 30% fewer parameters** (e.g., saves 184M parameters in the setting with ViT-L/14) and achieves comparable performance. Compared with Single-Task (LoRA), existing merging models methods perform much worse, while our Single-Task (LoRA) is the only one that can achieve comparable performance.
>
> ---
>
> > Q5. "The proposed approach appears to have a broad compatibility with several existing model merging methods, including Fisher-Merging (Matena \& Raffel, 2022), RegMean (Jin et al., 2023), and TIES-Merging (Yadav et al., 2023). It would be insightful to understand how the performance of the presented method fares when composed with these model merging methods."
>
> **A5.** The proposed PERU-FFT is **general and can be combined with any existing merging models methods**. In the submission, we combine PERU-FFT method with Task-Arithmetic. Following the reviewer's suggestions, we conducted additional experiments using the setting with ViT-B/32 to verify the compatibility of PERU-FFT with existing merging models methods. Table below shows the testing accuracy. As we can see, PERU-FFT is **consistently beneficial to existing methods** (Task-Arithmetic, Fisher-Merging, RegMean, TIES-Merging).
>
> We deeply appreciate the reviewer's suggestion and **have added the compatibility of our PERU-FFT as one of the main contributions** in the updated paper (Abstract, Conclusion, Section 4.3, Appendix B.1).
>
> \begin{array}{l|cccccccccc}
> \hline
> \text{Method} & \text{\\#params (M)} & \text{MNI} & \text{GTS} & \text{SVH} & \text{RES} & \text{SUN} & \text{EUR} & \text{DTD} & \text{CAR} & \text{Avg} \newline
> \hline
> \text{Task-Arithmetic} & 113 & 93.27 & 65.99 & 71.62 & 71.57 & 63.63 & 78.41 & 51.76 & 61.50 & 69.72 \newline
> \text{Task-Arithmetic + PERU-FFT } (1\\%) & 123 & 96.17 & 76.33 & 79.27 & 78.03 & 66.88 & 84.89 & 58.03 & 65.99 & 75.70 \newline
> \text{Task-Arithmetic + PERU-FFT } (5\\%) & 159 & 99.12 & 92.66 & 91.86 & 88.48 & 71.35 & 94.85 & 67.77 & 73.08 & 84.90 \newline
> \text{Task-Arithmetic + PERU-FFT } (10\\%) & 204 &  {\bf 99.49} &  {\bf 97.57} & {\bf 95.92} & {\bf 93.00} & {\bf 73.52} & {\bf 97.63} & {\bf 72.98} & {\bf 76.92} & {\bf 88.38} \newline
> \hline
> \text{Fisher-Merging} & 113 & 80.71 & 75.15 & 74.08 & 70.24 & 65.25 & 81.48 & 49.84 & 62.90 & 69.96 \newline
> \text{Fisher-Merging + PERU-FFT } (1\\%) & 123 & 92.29 & 69.23 & 71.50 & 77.54 & 68.27 & 79.59 & 56.28 & 67.40 & 72.76\newline
> \text{Fisher-Merging + PERU-FFT } (5\\%) & 159 & 98.81 & 91.39 & 90.31 & 88.73 & 72.38 & 94.33 & 67.18 & 74.26 & 84.67 \newline
> \text{Fisher-Merging + PERU-FFT } (10\\%) & 204 & {\bf 99.46} & {\bf 97.26} & {\bf 95.73} & {\bf 93.46} & {\bf 74.43} & {\bf 97.37} & {\bf 73.72} & {\bf 77.81} & {\bf 88.66} \newline
> \hline
> \text{RegMean} & 113 & 92.55 & 65.12 & 75.48 & 75.56 & 65.72 & 84.33 & 56.01 & 64.54 & 72.41 \newline
> \text{RegMean + PERU-FFT } (1\\%) & 123 & 93.79 & 71.46 & 78.77 & 77.95 & 67.47 & 87.15 & 58.14 & 66.47 & 75.15 \newline
> \text{RegMean + PERU-FFT } (5\\%) & 159 & 98.59 & 91.83 & 91.74 & 88.65 & 72.01 & 96.19 & 67.98 & 73.77 & 85.10\newline
> \text{RegMean + PERU-FFT } (10\\%) & 204 & {\bf 99.37} & {\bf 97.39} & {\bf 95.76} & {\bf 93.56} & {\bf 74.37} & {\bf 97.89} & {\bf 74.04} & {\bf 77.09} & {\bf 88.68}\newline
> \hline
> \text{TIES-Merging} & 113 & 97.79 & 75.30 & 84.10 & 70.71 & 59.24 & 75.89 & 53.51 & 58.72 & 71.91 \newline
> \text{TIES-Merging + PERU-FFT} (1\\%) & 123 &  98.82 & 86.25 & 87.27 & 75.79 & 61.29 & 87.15 & 58.78 & 62.22 & 77.20 &  \newline
> \text{TIES-Merging + PERU-FFT} (5\\%) & 159 & 99.44 & 96.42 & 94.52 & 86.86 & 67.01 & 95.22 & 67.45 & 70.41 & 84.67\newline
> \text{TIES-Merging + PERU-FFT} (10\\%) & 204 & {\bf 99.65} & {\bf 98.22} & {\bf 96.40} & {\bf 91.86} & {\bf 70.32} & {\bf 97.22} & {\bf 72.45} & {\bf 74.12} & {\bf 87.53}\newline
> \hline
> \end{array}

---

> ### Author Response · Authors · 2023-11-21
> **Reply to Reviewer a2T8 (4/4)**
>
> > Q6. "The supplementary material provided appears to be incorrect. It seems the authors mistakenly included the main text within the supplementary section.'"
>
> **A6.** We apologize for the confusion caused by supplementary material. Actually, this work does not have supplementary material and we mistakenly clicked the paper as supplementary material (which cannot be deleted anymore in the submission system).

---

> > ### Comment · Reviewer_a2T8 · 2023-11-22
> > **Response to authors rebuttal**
> >
> > Thank you for the informative rebuttal from the authors. However, I still have concerns regarding PERU-FFT. Specifically, it appears that the advantages of PERU-FFT over post-pruning become less significant as the number of parameters increases.

---

> > > ### Author Response · Authors · 2023-11-22
> > > **Reply to further comments**
> > >
> > > Thanks for your further comments.
> > >
> > > We deeply appreciate that **our previous reply has addressed all six concerns raised in your initial review**.
> > >
> > > For the new concern, we address it as follows.
> > >
> > > > Q7. However, I still have concerns regarding PERU-FFT. Specifically, it appears that the advantages of PERU-FFT over post-pruning become less significant as the number of parameters increases.
> > >
> > > **A7.** When merging models into a multi-task model, we expect to obtain a merged model with **fewer additional parameters** which can achieve satisfactory performance. When keeping a **small** number of parameters, PERU-FFT consistently **performs much better** than post-pruning, as shown in Figure 7 of the paper. Specifically, as shown in the table below, PERU-FFT **has much higher accuracy** (averaged over eight tasks) than post-pruning **by a large margin** when keeping top-1\% or top-5\% parameters. If we continue to increase the number of retained parameters, both PERU-FFT and post-pruning will **saturate and reach Single-Task performance**, as shown in Figure 7 of the paper.
> > >
> > >
> > > \begin{array}{lcccccc}
> > > \hline
> > > \text{Method} &   \text{ViT-B/32} & \text{ViT-B/16} & \text{ViT-L/14} \newline
> > > \hline
> > > \text{Post-Pruning (1\\%)} & 55.14 & 62.90 & 72.54 \newline
> > > \text{PERU-FFT (1\\%)} & {\bf 75.70} & {\bf 80.23} & {\bf 87.21}\newline
> > > \hline
> > > \text{Post-Pruning (5\\%)} & 78.12 & 82.13 & 87.52  \newline
> > > \text{PERU-FFT (5\\%)}& {\bf 84.90} & {\bf 87.97} & {\bf 91.20} \newline
> > > \hline
> > > \end{array}
> > >
> > > Please let us know if you have any remaining questions; we are more than happy to address them.

---

> > > > ### Comment · Reviewer_a2T8 · 2023-11-22
> > > > **Response to authors rebuttal**
> > > >
> > > > Thank you for your comprehensive response. The authors have successfully addressed most of my concerns, which is greatly appreciated. In light of these clarifications, I am inclined to revise my evaluation and increase my score to a 6.

---

> > > > > ### Author Response · Authors · 2023-11-22
> > > > > **Thank you!**
> > > > >
> > > > > We are glad that our reply addressed your concerns.
> > > > >
> > > > > We sincerely thank you again for the suggested experiments, which greatly improved our work.
> > > > >
> > > > > Thanks for raising the score!

---

### Official Review · Reviewer_tpTw · 2023-10-30

**Soundness:** 2 fair
**Presentation:** 3 good
**Contribution:** 2 fair
**Rating:** 5
**Confidence:** 4

**Summary:**

The paper targets the task of merging multiple fine-tuned task-specific models into a single multi-task model in a training-free manner. For full fine-tuned models, authors inject a sparse task vector into the merged model by magnitude pruning. For LoRA fine-tuned models, authors use a lower-rank matrix to approximate the LoRA matrix by singular value decomposition. By adding a small number of task-specific weights to the merged model, the merged model shows significant improvement over existing methods.

**Strengths:**

1. The writing is clear and easy to read.
2. The performance improvement is significant.
3. The method is evaluated on both CV and NLP with different backbone sizes.

**Weaknesses:**

1. The paper targets the task of merging multiple fine-tuned task-specific models into a single multi-task model. However, why is this problem important? What's the practical value and potential impact in real-world scenarios? Could you give a specific real-world example where a user needs such a merge model? Model users are typically interested in the performance of their tasks. If the user needs to deal with multiple tasks, it's not very likely that some fine-tuned models online can directly solve their problems unless the fine-tuned data are almost the same as the users.

2.  TIES-Merging has shown that keeping only the top-20% of the parameter based on magnitude does not degrade the performance too much. The proposed method just adds the top-m% parameters to the merged model. I think it's quite incremental and the contribution is marginal.

3. Typically, users need a multi-task model to handle data in a similar domain. The current benchmark is so diverse, with handwritten digits, remote-sensing, cars etc datasets. It would be great to see the performance of the merging algorithm on similar datasets. Could the authors provide merging performance within the Natural, Specialized, and Structured groups in the VTAB dataset?

4. Given the prevalent use of CNN-based models like ResNet in various domains, it is crucial to understand how well the proposed PERU-FFT method generalizes to these architectures. Including experiments or discussions on this aspect would broaden the paper's applicability and appeal to a wider audience.


5. Because PERU adds task-specific weight to the model, I think it would be great to compare it with multi-task training performance, where the multi-task model is trained on all data with a specific head for each task.

6. The authors did not mention the limitations of their method and potential future work. The paper does not explore or discuss potential failure cases of the proposed methods. Understanding when and why the methods might fail is crucial for practical applications.

**Questions:**

Please refer to the Weaknesses section.

---

> ### Author Response · Authors · 2023-11-21
> **Reply to Reviewer tpTw (1/4)**
>
> Thanks for your thoughtful review and valuable feedback. We address your concerns as follows.
>
> > Q1. "The paper targets the task of merging multiple fine-tuned task-specific models into a single multi-task model. However, why is this problem important? What's the practical value and potential impact in real-world scenarios? "
>
> **A1.** **Importance and practical value of Merging models.**
>
> (i) The motivation for reusing finetuned task-specific models is clear. As discussed in the Introduction of the submission, pre-trained large-scale models (e.g., ViT, T5, LLaMA) are powerful nowadays, and many task-specific models are provided online for public use (e.g., by 2023, more than 120, 000 models are on https://huggingface.co/). In real-world applications, when we need to solve multiple tasks, it is plausible to use some finetuned models available online to **save the cost of labeling data and training models**. Note that finetuning a large pre-trained model is very expensive and not every company/user can afford it. **Reusing existing finetuned models is training-free and can reduce CO2 emissions caused by training models**. This research topic is practically important and many recent attempts are proposed, e.g., Fisher-Merging (NeurIPS 2021), Task-Arithmetic (ICLR 2023), RegMean (ICLR 2023), TIES-Merging (NeurIPS 2023)
>
> (ii) Merging models can combine several models with different functions into a merged model **without requiring task data**. Many companies/users are willing to share their model checkpoints online but **do not want to release the training data for privacy concerns**. In this case, merging models is an option of combining various models in a multi-task model, while re-training on task data is infeasible.
>
> (iii) Merging models can **add new knowledge** (task-specific models) into a merged model without further training and data annotation/collection.
>
> ---
>
> > Q2. "Could you give a specific real-world example where a user needs such a merge model?"
>
> **A2.** `Real-World Example 1.` If NLP users want to build a multi-task model (sentiment classification, detecting semantical equivalence between two sentences, detecting grammatical correctness of sentence), they can merge the following task-specific models (checkpoints) finetuned from the `T5-base` model:
>
> - sentiment classification: https://huggingface.co/PavanNeerudu/t5-base-finetuned-sst2
> - detecting semantical equivalence: https://huggingface.co/PavanNeerudu/t5-base-finetuned-mrpc
> - detecting grammatical correctness: https://huggingface.co/PavanNeerudu/t5-base-finetuned-cola
>
> Also, they can merge the task-specific models finetuned from the more powerful `GPT-2` model:
>
> - sentiment classification: https://huggingface.co/PavanNeerudu/gpt2-finetuned-sst2
> - detecting semantical equivalence: https://huggingface.co/PavanNeerudu/gpt2-finetuned-mrpc
> - detecting grammatical correctness: https://huggingface.co/PavanNeerudu/gpt2-finetuned-cola
>
> `Real-World Example 2.` To build a **powerful** language model capable of following instructions in **multiple languages** (e.g., Simplified Chinese, Traditional Chinese, Japanese, Korean), one can merge the following models fine-tuned from LLaMA-2 (7B):
> - `Simple Chinese`: https://huggingface.co/LinkSoul/Chinese-Llama-2-7b
> - `Traditional Chinese`: https://huggingface.co/yentinglin/Taiwan-LLM-7B-v2.0.1-chat
> - `Japanese`: https://huggingface.co/elyza/ELYZA-japanese-Llama-2-7b
> - `Korean`: https://huggingface.co/quantumaikr/KoreanLM-llama-2-7B-finetuned
>
> ---
>
> > Q3. "Model users are typically interested in the performance of their tasks. If the user needs to deal with multiple tasks, it's not very likely that some fine-tuned models online can directly solve their problems unless the fine-tuned data are almost the same as the users."
>
> **A3.** Indeed, previous merging algorithms and ours **can also merge task-specific models that users finetune on their data**. Users/organizations/companies can finetune models and share models with each other to merge into a powerful multitask model. We know that sharing data is another option for training a multitask model, but many users/organizations/companies are unwilling to do that due to privacy concerns.

---

> ### Author Response · Authors · 2023-11-21
> **Reply to Reviewer tpTw (2/4)**
>
> > Q4. "TIES-Merging has shown that keeping only the top-20% of the parameter based on magnitude does not degrade the performance too much. The proposed method just adds the top-m% parameters to the merged model. I think it's quite incremental and the contribution is marginal."
>
> **A4.**
> (i) TIES-Merging adds the top-20% of $\theta_t-\theta_0$ (denoted $\hat{\bf v}_t$) to the pre-trained model $\theta_0$ (i.e., Post-Pruning in Algorithm 1 of the submission), thus, the pruned task model $\theta_0 + \hat{\bf v}_t$ **does not use any shared knowledge from other tasks**. In contrast, our PERU-FFT adds the top-20\% of $\theta_t - \theta^\star$ to the merged model $\theta^\star$, which **contains knowledge from other tasks**. PERU-FFT is simple but effective. As shown in Figure 1 of the submission, **PERU-FFT consistently performs better than Post-Pruning**, demonstrating that **sharing knowledge across tasks is more effective**.
>
> (ii) Post-Pruning introduced in TIES-Merging is **not suitable** for merging LoRA-finetuned models as it has **NOT** leveraged the special structure ${\bf A}_t{\bf B}_t^\top$ in task vectors. As shown in Figure 2, TIES-Merging is much worse than Single-Task in all three settings. We propose a novel method **PERU-LoRA** to merge LoRA finetuned models by using **singular value decomposition** to compress the LoRA matrices. As can be seen from Figure 2, **PERU-LoRA consistently has a much higher accuracy than TIES-Merging** (e.g., +25\% on ViT-B/32, +20\% on ViT-B/16, +17\% on ViT-L/14).
>
> Our **main** contribution is proposing **two novel methods PERU-FFT and PERU-LoRA** for merging fully and LoRA finetuned models, which perform much better than existing merging models methods (including TIES-Merging).
>
> ---
>
> > Q5. "Typically, users need a multi-task model to handle data in a similar domain. The current benchmark is so diverse, with handwritten digits, remote-sensing, cars etc datasets. It would be great to see the performance of the merging algorithm on similar datasets. Could the authors provide merging performance within the Natural, Specialized, and Structured groups in the VTAB dataset?"
>
> **A5.** Thanks for your insightful suggestions. We conducted an additional experiment on the *Natural* group of *VTAB* [1] using *ViT-B/16*. Table below shows the testing accuracy. The observations are **consistent** with the experimental results in Section 4.1 of the submission. Specifically, as we can see, by keeping top-10% values, both PERU-FFT and Post-Pruning achieve comparable performance with Single-Task, but are more parameter-efﬁcient (4.5$\times$ fewer parameters). PERU-FFT (with additional 1% parameters per task) performs better than the existing merging models method, showing the effectiveness of introducing sparse task-speciﬁc vectors into the merged model.
> Compared with Post-Pruning, PERU-FFT achieves higher accuracy (averaged over seven tasks), showing that merging the task-speciﬁc models before pruning the task vectors is more effective. We added the results in Appendix B.4 of the updated paper.
>
> \begin{array}{lccccc}
> \hline
> & \text{\\#params (M)}
> & \text{CIF} & \text{CAL} & \text{DTD} & \text{FLO} & \text{PET} & \text{SVH} & \text{SUN} & \text{Avg}
> \newline
> \hline
> \text{Pre-Trained} & 112 & 66.91 & 82.76 & 45.11 & 71.33 & 87.19 & 51.98 & 65.50 & 67.25 \newline
> \hline
> \text{Single-Task} & 894 & 90.23 & 97.20 & 82.29 & 94.88 & 94.55 & 97.86 & 78.71 & 90.82  \newline
> \hline
> \text{Task-Arithmetic} & 112 & 83.33 & 87.07 & 52.87 & 66.87 & 89.10 & 83.52 & 67.01 & 75.68  \newline
> \text{Fisher-Merging} &112 & 80.81 & 86.96 & 51.28 & 73.72 & 89.51 & 69.92 & 69.32 & 74.50 & \newline
> \text{RegMean} & 112 & 81.32 & 87.07 & 55.53 & 75.41 & 90.38 & 84.94 & 69.88 & 77.79 \newline
> \text{TIES-Merging} & 112 & 82.77 & 87.69 & 57.39 & 70.39 & 89.59 & 88.28 & 67.42 & 77.65 \newline
> \hline
> \text{Post-Pruning ($1\\%$)} & 121 & 74.37 & 85.06 & 49.63 & 73.59 & 88.23 & 60.53 & 68.51 & 71.42 \newline
> \text{Post-Pruning ($5\\%$)} & 157 & 86.55 & 90.60 & 64.89 & 78.45 & 91.58 & 82.06 & 74.41 & 81.22 \newline
> \text{Post-Pruning ($10\\%$)} & 201& 89.61 & 93.68 & 76.01 & {\bf 84.34} & {\bf 94.41} & 94.42 & {\bf 77.00} & 87.07 \newline
> \hline
> \text{PERU-FFT ($1\\%$)} & 121& 85.39 & 89.93 & 58.30 & 70.89 & 91.61 & 87.78 & 69.76 & 79.09 \newline
> \text{PERU-FFT ($5\\%$)} & 157& 88.50 & 92.56 & 69.73 & 77.62 & 93.59 & 94.71 & 74.18 & 84.41 \newline
> \text{PERU-FFT ($10\\%$)} &201 & {\bf 89.75} & {\bf 93.90} & {\bf 76.22} & 83.98 & 94.06 & {\bf 97.01} & 76.46 & {\bf 87.34} \newline
> \hline
> \end{array}

---

> ### Author Response · Authors · 2023-11-21
> **Reply to Reviewer tpTw (3/4)**
>
> > Q6. "Given the prevalent use of CNN-based models like ResNet in various domains, it is crucial to understand how well the proposed PERU-FFT method generalizes to these architectures. Including experiments or discussions on this aspect would broaden the paper's applicability and appeal to a wider audience."
>
> **A6.** Thanks for your insightful suggestions. We conducted an additional experiment using a CNN-based model *ConvNeXt-Base* [2] on eight computer vision tasks. Table below shows the testing accuracy. The following observations are **consistent** with the results of ViT-based models in Section 4.1 of the submission. As we can see, (i) compared with Single-Task method, both PERU-FFT (10%) and Post-Pruning (10%) achieve comparable performance but have 4.5$\times$ fewer parameters. With an additional 1% of parameters per task, PERU-FFT performs better than the existing merging models method, showing that introducing a sparse task-speciﬁc vector into the merged model is effective. PERU-FFT consistently achieves higher accuracy (averaged over seven tasks) than Post-Pruning, showing that merging the task-speciﬁc models before pruning the task vectors is more effective. We added the results in Appendix B.5 of the updated paper.
>
> \begin{array}{lcccccccc}
> \hline
> \text{Method} & \text{\\#params (M)}
> & \text{MNI} & \text{GTS} & \text{SVH} & \text{RES} & \text{SUN} & \text{EUR} & \text{DTD} & \text{CAR} & \text{Avg} \newline
> \hline
> \text{Pre-Trained} & 179 & 64.39 & 46.56 & 53.73 & 65.94 & 71.61 & 52.37 & 61.54 & 91.24 & 63.42\newline
> \hline
> \text{Single-Task}  & 1,435 & 99.78 & 99.22 & 98.01 & 96.67 & 80.49 & 99.19 & 85.74 & 94.91 & 94.25\newline
> \hline
> \text{Task-Arithmetic} & 179 & 97.73 & 81.31 & 82.96 & 76.56 & 72.12 & 78.07 & 68.40 & 92.87 & 81.25\newline
> \text{Fisher-Merging} & 179 & 95.47 & 67.67 & 77.93 & 76.21 & 72.80 & 74.22 & 67.82 & 92.53 & 78.08\newline
> \text{RegMean} & 179 & 97.92 & 81.25 & 86.47 & 80.65 & 74.00 & 89.26 & 72.55 & 93.82 & 84.49\newline
> \text{TIES-Merging} & 179 & 99.16 & 85.32 & 88.83 & 72.97 & 69.44 & 78.37 & 65.27 & 91.93 & 81.41\newline
> \hline
> \text{Post-Pruning ($1\\%$)} & 194 & 85.64 & 54.60 & 65.75 & 73.29 & 73.68 & 65.30 & 67.13 & 92.54 & 72.24\newline
> \text{Post-Pruning ($5\\%$)} & 251 & 99.04 & 83.80 & 89.97 & 87.43 & 77.26 & 91.11 & 76.33 & 94.28 & 87.40\newline
> \text{Post-Pruning ($10\\%$)} & 323 & 99.65 & 95.65 & 96.47 & 93.21 & {\bf 79.29} & {\bf 97.89} & 80.74 & 94.86 & 92.22\newline
> \hline
> \text{PERU-FFT ($1\\%$)} & 194 & 99.02 & 88.49 & 87.86 & 81.89 & 73.79 & 86.26 & 72.07 & 93.56 & 85.37\newline
> \text{PERU-FFT ($5\\%$)} & 251 & 99.60 & 95.90 & 95.01 & 90.08 & 76.57 & 95.00 & 78.09 & 94.68 & 90.62\newline
> \text{PERU-FFT ($10\\%$)} & 323 & {\bf 99.72} & {\bf 98.02} & {\bf 97.33} & {\bf 93.65} & 78.66 & 97.41 & {\bf 82.39} & {\bf 95.04} & {\bf 92.78} \newline
> \hline
> \end{array}

---

> ### Author Response · Authors · 2023-11-21
> **Reply to Reviewer tpTw (4/4)**
>
> > Q7. "Because PERU adds task-specific weight to the model, I think it would be great to compare it with multi-task training performance, where the multi-task model is trained on all data with a specific head for each task."
>
> **A7.** As suggested, we compare the performance of merging models with the multi-task learning method (MTL) using ViT-B/32, where the result is reported in Task-Arithmetic (https://github.com/mlfoundations/task_vectors/issues/5). Table below shows the testing accuracy. We can see that PERU-FFT ${\bf u}_t$ (10\%) achieves **comparable** performance with MTL.
> Note that comparing MTL to the merged model approach is **not an apples-to-apples comparison** as (i) MTL requires the **availability of all task data** for training while merging model methods use task-specific models available online and **do not need the task data**. (ii) Furthermore, MTL needs to **train the MTL model**, which is expensive for large-scale models. In contrast, merging model methods is **training-free**. We added the results in the updated paper.
>
> \begin{array}{lccccccccc}
> \hline
> \text{Method} & \text{\\#params (M)} & \text{MNI} & \text{GTS} & \text{SVH} & \text{RES} & \text{SUN} & \text{EUR} & \text{DTD} & \text{CAR} & \text{Avg} \newline
> \hline
> \text{Pre-Trained} & 113 & 48.25 & 32.56 & 31.61 & 60.65 & 63.18 & 45.11 & 43.99 & 59.74 & 48.14 \newline
> \hline
> \text{Single-Task}  & 908 & 99.72 & 99.23 & 97.42 & 95.56 & 75.03 & 99.00 & 79.47 & 78.73 & 90.52 \newline
> \text{MTL}  & 113 & 99.45 & 98.91 & 95.80 & 93.90 & 72.85 & 98.22 & 77.87 & 74.44 & 88.93  \newline
> \hline
> \text{Task-Arithmetic} & 113 & 93.27 & 65.99 & 71.62 & 71.57 & 63.63 & 78.41 & 51.76 & 61.50 & 69.72 \newline
> \text{Fisher-Merging} & 113 & 80.71 & 75.15 & 74.08 & 70.24 & 65.25 & 81.48 & 49.84 & 62.90 & 69.96 \newline
> \text{RegMean} & 113 & 92.55 & 65.12 & 75.48 & 75.56 & 65.72 & 84.33 & 56.01 & 64.54 & 72.41 \newline
> \text{TIES-Merging} & 113 & 97.79 & 75.30 & 84.10 & 70.71 & 59.24 & 75.89 & 53.51 & 58.72 & 71.91 \newline
> \hline
> \text{Post-Pruning ${\bf v}_t$} (1\\%) & 123 & 58.41 & 40.61 & 39.38 & 67.08 & 66.63 & 56.26 & 48.83 & 63.95 & 55.14 \newline
> \text{Post-Pruning ${\bf v}_t$} (5\\%) & 159 & 95.82 & 78.61 & 74.35 & 83.67 & 71.60 & 85.81 & 62.39 & 72.73 & 78.12 \newline
> \text{Post-Pruning ${\bf v}_t$} (10\\%) & 204 & 99.17 & 95.30 & 93.85 & 92.13 & {\bf 74.39} & 96.37 & 71.97 &  {\bf 77.09}  & 87.53 \newline
> \hline
> \text{PERU-FFT ${\bf u}_t$} (1\\%) & 123 & 96.17 & 76.33 & 79.27 & 78.03 & 66.88 & 84.89 & 58.03 & 65.99 & 75.70 \newline
> \text{PERU-FFT ${\bf u}_t$} (5\\%) & 159 & 99.12 & 92.66 & 91.86 & 88.48 & 71.35 & 94.85 & 67.77 & 73.08 & 84.90 \newline
> \text{PERU-FFT ${\bf u}_t$} (10\\%) & 204 &  {\bf 99.49} &  {\bf 97.57} & {\bf 95.92} & {\bf 93.00} & 73.52 & {\bf 97.63} & {\bf 72.98} & 76.92 & {\bf 88.38} \newline
> \hline
> \end{array}
>
> ---
>
> > Q8. "The authors did not mention the limitations of their method and potential future work. The paper does not explore or discuss potential failure cases of the proposed methods. Understanding when and why the methods might fail is crucial for practical applications."
>
> **A8.** Like existing model merging methods (Task-Arithmetic, Fisher-Merging, RegMean, TIES-Merging), the proposed PERU is only suitable for task-specific models with the same architecture. Another limitation is that PERU requires the task-specific models to be finetuned from the same pre-trained model, which is also necessary for existing model merging methods. Weakening these assumptions is an important research direction in merging models. We added the limitations and future works in Appendix A of the updated paper.
>
> ### **References**
>
> [1] Xiaohua Zhai, Joan Puigcerver, Alexander Kolesnikov, Pierre Ruyssen, Carlos Riquelme, Mario Lucic, Josip Djolonga, Andre Susano Pinto, Maxim Neumann, Alexey Dosovitskiy, Lucas Beyer, Olivier Bachem, Michael Tschannen, Marcin Michalski, Olivier Bousquet, Sylvain Gelly, and Neil Houlsby. A large-scale study of representation learning with the visual task adaptation benchmark. Preprint arXiv:1910.04867, 2019.
>
> [2] Zhuang Liu, Hanzi Mao, Chao-Yuan Wu, Christoph Feichtenhofer, Trevor Darrell, and Saining Xie. A ConvNet for the 2020s. In IEEE/CVF Conference on Computer Vision and Pattern Recognition, 2022.

---

> ### Author Response · Authors · 2023-11-22
> **Are there any remaining concerns?**
>
> Dear Reviewer tpTw,
>
> We would like to thank you again for your detailed reviews and suggested experiments to improve this work. We have conducted all required experiments and the results are in the above reply. We hope that our reply has satisfactorily addressed your concerns.
>
> If there is any additional explanation or experiments that can save the reviewer’s time to understand our paper and clarify the concerns, we will be more than happy to do so.
>
> Best,
>
> Authors

---

> ### Author Response · Authors · 2023-11-23
> **Would you mind checking our responses? (waiting for your feedback)**
>
> Dear Reviewer tpTw,
>
> We again express our deep gratitude for your time and efforts in our work.
>
> As the author-reviewer discussion period is coming to the end, please let us know if you have any further concerns or questions; we will be happy to address them.
>
> Best,
>
> Authors

---

### Official Review · Reviewer_wHfz · 2023-10-31

**Soundness:** 3 good
**Presentation:** 3 good
**Contribution:** 2 fair
**Rating:** 6
**Confidence:** 3

**Summary:**

Utilizing distinct fine-tuned models for different tasks imposes a significant burden on storage and servicing requirements. Current merging methods exhibit a significant performance disparity when compared to single-task models. This work proposes a Parameter-Efficient method for ReUsing fine-tuned models (PERU), which enhances by employing pruning on fully fine-tuned models and implementing Singular Value Decomposition (SVD) on LoRA fine-tuned models. The experimental result demonstrates that PERU outperforms existing merging methods and achieves comparable performance compared to single-task fine-tuned models.

**Strengths:**

1) The use of pruning and SVD on merging fine-tuned models is simple but effective.
2) The result is a significant improvement over the previous methods.

**Weaknesses:**

1) According to Algorithm 1 and Algorithm 2, the PERU method still uses a specific model for a specific task, which is different from previous merging methods that use the same parameters for all tasks. In the paper, you mention that merging task models into a shared one will cause a parameter inference problem, which affects the performance. While PERU circumvents this issue, it introduces a new challenge by employing task-specific parameters. Consequently, direct comparisons between PERU and other methods may not be entirely equitable.

2) The absence of ablation experiments utilizing only a subset of tasks. As the number of tasks increases, will the performance of your method be affected?

**Questions:**

1) Do previous methods typically require the specification of dataset sources when addressing different classification tasks? If not, is your method capable of performing similarly without declaring the dataset origins for each task?

2) With a limited scope of only two or four tasks, does your proposed method continue to significantly outperform existing merging methods?

3) What is the result if we don’t prune for $v_t$ or $u_t$ and we direct prune $\theta_t$?

---

> ### Author Response · Authors · 2023-11-21
> **Reply to Reviewer wHfz (1/2)**
>
> Thanks for your thoughtful review and valuable feedback. We address your concerns as follows.
>
> > Q1. "According to Algorithm 1 and Algorithm 2, the PERU method still uses a specific model for a specific task, which is different from previous merging methods that use the same parameters for all tasks. In the paper, you mention that merging task models into a shared one will cause a parameter inference problem, which affects the performance. While PERU circumvents this issue, it introduces a new challenge by employing task-specific parameters. Consequently, direct comparisons between PERU and other methods may not be entirely equitable."
>
> **A1.** Our PERU methods propose to use a shared model and a set of pruned task-specific vectors. **A few additional parameters** in PERU are **affordable** (e.g., less than 100M when using ViT-B/32), **contributing a large improvement** over previous merging methods, as shown in Figure 1 of the submission. For example, using ViT-B/32, PERU-FFT (10%) achieves much higher accuracy (averaged over eight tasks) than all existing merging methods (+16\%), suggesting a small number of task-specific parameters is valuable in practice. Compared with Single-Task, which is expensive in storing all task models), PERU-FFT is parameter-efficient and achieves comparable performance.
>
>
>
> ---
>
> > Q2. "The absence of ablation experiments utilizing only a subset of tasks. As the number of tasks increases, will the performance of your method be affected?"
> >
> > "With a limited scope of only two or four tasks, does your proposed method continue to significantly outperform existing merging methods?"
>
> **A2.** As suggested, we conducted an ablation experiment using **four** computer vision tasks with *ViT-B/32*. Tables below show the testing accuracy for fully-FT and LoRA-FT settings. As can be seen, for merging fully finetuned models, with a small number of parameters, Post-Pruning (5%) and Post-Pruning (10%) **outperform** existing merging methods (Task-Arithmetic, Fisher-Merging, RegMean, TIES-Merging), demonstrating the effectiveness of introducing sparse task-speciﬁc vectors into the shared model. Compared with Post-Pruning, PERU-FFT achieves higher accuracy (averaged over four tasks), suggesting that merging the task-speciﬁc models before pruning the task vectors is more effective. Moreover, compared with Single-Task method, PERU-FFT (10\%) is more parameter-efficient and achieves comparable performance. For merging LoRA finetuned models, PERU-LoRA **achieves higher accuracy** (averaged over four tasks) than previous merging methods. We added the results in Appendix B.3 of the updated paper.
>
>
>
> `Fully-FT`
>
> \begin{array}{lcccccc}
> \hline
> \text{Method} & \text{\\#params (M)} & \text{SVH} & \text{EUR} & \text{DTD} & \text{CAR} & \text{Avg} \newline
> \hline
>  \text{Pre-Trained} & 113 & 31.61 & 45.11 & 43.99 & 59.74 & 45.11 \newline
> \hline
>  \text{Single-Task} & 452 & 97.42 & 99.00 & 79.47 & 78.73 & 88.66 \newline
>  \hline
> \text{Task-Arithmetic} & 113 & 77.30 & 92.78 & 61.49 & 67.83 & 74.85\newline
> \text{Fisher-Merging} &113& 72.02 & 89.89 & 58.72 & 67.50 & 72.03  \newline
> \text{RegMean} &113& 86.34 & 95.78 & 64.41 & 69.11 & 78.91 \newline
> \text{TIES-Merging} &113 & 88.95 & 94.11 & 66.70 & 70.68 & 80.11  \newline
> \hline
> \text{Post-Pruning (1\\%)} & 118& 39.39 & 56.26 & 48.78 & 63.95 & 52.05 \newline
> \text{Post-Pruning (5\\%)} & 136 & 74.35 & 85.81 & 62.39 & 72.74 & 73.83 \newline
> \text{Post-Pruning (10\\%)} & 159 & 93.85 & 96.37 & 71.91 & 77.09 & 84.81 \newline
> \hline
> \text{PERU-FFT (1\\%)} &118& 84.48 & 95.04 & 64.95 & 70.87 & 78.84 \newline
> \text{PERU-FFT (5\\%)} & 136 & 94.45 & 97.41 & 71.81 & 75.69 & 84.84\newline
> \text{PERU-FFT (10\\%)} & 159 & {\bf 96.68} & {\bf 98.15} & {\bf 75.53} & {\bf 77.89} & {\bf 87.06} \newline
> \hline
> \end{array}
>
> `LoRA-FT`
>
> \begin{array}{lcccccc}
> \hline
> \text{Method} & \text{\\#params (M)} & \text{SVH} & \text{EUR} & \text{DTD} & \text{CAR} & \text{Avg} \newline
> \hline
>  \text{Pre-Trained} & 113 & 31.61 & 45.11 & 43.99 & 59.74 & 45.11 \newline
> \hline
>  \text{Single-Task} & 153 & 97.34 & 98.63 & 76.91 & 77.25 & 87.52 \newline
>  \hline
> \text{Task-Arithmetic} &113& 62.16 & 80.78 & 53.35 & 64.08 & 65.09\newline
> \text{Fisher-Merging} &113& 71.94 & 89.81 & 58.72 & 67.54 & 72.01 \newline
> \text{RegMean} &113& 89.33 & 94.85 & 61.60 & 67.57 & 78.34 \newline
> \text{TIES-Merging} &113 & 47.16 & 67.52 & 48.30 & 62.21 & 56.29 \newline
> \hline
> \text{PERU-LoRA (4)} &114& 93.98 & 95.37 & 65.37 & 62.74 &79.37 \newline
> \text{PERU-LoRA (8)} &116& 96.45 & 98.26 & 72.55 & 67.35 & 83.66 \newline
> \text{PERU-LoRA (16)} &118& {\bf 97.08} & {\bf 98.37} & {\bf 76.44} & {\bf 71.31} & {\bf 85.80} \newline
> \hline
> \end{array}

---

> ### Author Response · Authors · 2023-11-21
> **Reply to Reviewer wHfz (2/2)**
>
> > Q3. "Do previous methods typically require the specification of dataset sources when addressing different classification tasks? If not, is your method capable of performing similarly without declaring the dataset origins for each task?""
>
> **A3.** The answer to the first question is **Yes**. Previous methods and ours **need to know** the specification of dataset sources when addressing different classification tasks. Indeed, for each sample, previous methods need to know its task identity and use the corresponding **task-specific classification head**. E.g.,
>
> - Task-Arithmetic: Line 17 in `src/eval.py`, see [https://github.com/mlfoundations/task_vectors/blob/main/src/eval.py#L17](https://github.com/mlfoundations/task_vectors/blob/main/src/eval.py#L17)
> - Fisher Merging: see the last paragraph (Unmergeable Parameters) in Section 2 of the paper ([https://openreview.net/pdf?id=LSKlp_aceOC](https://openreview.net/pdf?id=LSKlp_aceOC))
> - TIES-Merging follows the experimental setup use the model checkoints provided by Task-Arithmetic (see Appendix C.1 and the footnote on Page 21 of the TIES-Merging paper https://openreview.net/pdf?id=xtaX3WyCj1)
> - In addition, TIES-Merging and RegMean apply **task-specific templates** for samples in NLP tasks. Hence, they need to know the task identity for data points; see the *Merging PEFT Models* paragraph on Page 6 of TIES-Merging (https://openreview.net/pdf?id=xtaX3WyCj1), and Page 15 of RegMean (https://openreview.net/pdf?id=FCnohuR6AnM).
>
> This is a **limitation** of merging models methods and we added it in Appendix A of the updated paper.
>
> ---
>
> > Q4. "What is the result if we don’t prune for ${\bf v}_t$ or ${\bf u}_t$ and we direct prune $\theta_t$?"
>
> **A4.** As suggested, we conducted additional experiments with ViT-B/32 to compare the performance of pruning $\theta_t$ and pruning ${\bf v}_t$ or ${\bf u}_t$. Table below shows the testing accuracy, where Pruning $\theta_t$ (m\%) denotes keeping the top-m\% parameters in $\theta_t$. As can be seen, **pruning $\theta_t$ is not effective**. For example, **Pruning $\theta_t$ (50%) has very low accuracy. In contrast, keeping top-10% of ${\bf v}_t$ or ${\bf u}_t$ perform much better (+80\%)**. Compared with Pruning $\theta_t$ (90%), PERU-FFT ${\bf u}_t$ (10\%) achieves comparable performance but has $4\times$ fewer parameters. We added the results in Appendix B.6 of the updated paper.
>
> \begin{array}{lccccccccccc}
> \hline
> \text{Method} & \text{\\#params (M)} & \text{MNI} & \text{GTS} & \text{SVH} & \text{RES} & \text{SUN} & \text{EUR} & \text{DTD} & \text{CAR} & \text{Avg} \newline
> \hline
> \text{Pre-Trained} & 113 & 48.25 & 32.56 & 31.61 & 60.65 & 63.18 & 45.11 & 43.99 & 59.74 & 48.14 \newline
> \hline
> \text{Single-Task}  & 908 & 99.72 & 99.23 & 97.42 & 95.56 & 75.03 & 99.00 & 79.47 & 78.73 & 90.52 \newline
> \hline
> \text{Pruning $\theta_t$ } (10\\%) &91 & 9.82 & 0.70 & 8.49 & 3.00 & 0.25 & 9.52 & 1.60 & 0.60 & 4.25 \newline
> \text{Pruning $\theta_t$ } (20\\%) &181& 10.28 & 1.77 & 6.71 & 2.11 & 0.28 & 16.11 & 2.45 & 0.46 & 5.02 \newline
> \text{Pruning $\theta_t$ } (30\\%) &271& 9.91 & 3.72 & 17.62 & 2.63 & 0.44 & 10.78 & 2.23 & 0.49 & 5.98\newline
> \text{Pruning $\theta_t$ } (40\\%) &362& 10.09 & 5.08 & 6.29 & 4.48 & 0.32 & 14.48 & 2.71 & 0.40 & 5.48 \newline
> \text{Pruning $\theta_t$ } (50\\%) &452& 10.94 & 5.59 & 20.45 & 6.73 & 0.92 & 17.00 & 7.23 & 0.53 & 8.67\newline
> \text{Pruning $\theta_t$ } (60\\%) &542& 84.54 & 43.72 & 63.88 & 44.63 & 15.23 & 34.67 & 31.91 & 3.47 & 40.26  \newline
> \text{Pruning $\theta_t$ } (70\\%) & 632 & 98.83 & 80.37 & 91.32 & 77.48 & 48.49 & 70.11 & 56.22 & 38.75 & 70.20\newline
> \text{Pruning $\theta_t$ } (80\\%) & 723 & 99.55 & 95.04 & 96.35 & 88.70 & 64.13 & 87.81 & 72.18 & 65.24 & 83.63 \newline
> \text{Pruning $\theta_t$ } (90\\%) & 814 & 99.69 & 99.06 & 97.39 & 95.24 & 73.59 & 98.81 & 79.10 & 77.08 & 89.99\newline
> \hline
> \text{Post-Pruning ${\bf v}_t$} (1\\%) & 123 & 58.41 & 40.61 & 39.38 & 67.08 & 66.63 & 56.26 & 48.83 & 63.95 & 55.14 \newline
> \text{Post-Pruning ${\bf v}_t$} (5\\%) & 159 & 95.82 & 78.61 & 74.35 & 83.67 & 71.60 & 85.81 & 62.39 & 72.73 & 78.12 \newline
> \text{Post-Pruning ${\bf v}_t$} (10\\%) & 204 & 99.17 & 95.30 & 93.85 & 92.13 & 74.39 & 96.37 & 71.97 &  77.09  & 87.53 \newline
> \hline
> \text{PERU-FFT ${\bf u}_t$ } (1\\%) & 123 & 96.17 & 76.33 & 79.27 & 78.03 & 66.88 & 84.89 & 58.03 & 65.99 & 75.70 \newline
> \text{PERU-FFT ${\bf u}_t$ } (5\\%) & 159 & 99.12 & 92.66 & 91.86 & 88.48 & 71.35 & 94.85 & 67.77 & 73.08 & 84.90 \newline
> \text{PERU-FFT ${\bf u}_t$ } (10\\%) & 204 &  99.49 &  97.57 & 95.92 & 93.00 & 73.52 & 97.63 & 72.98 & 76.92 & 88.38 \newline
> \hline
> \end{array}

---

> > ### Author Response · Authors · 2023-11-23
> > **Would you mind checking our responses? (waiting for your feedback)**
> >
> > Dear Reviewer wHfz,
> >
> > We again express our deep gratitude for your time and efforts in our work.
> >
> > As the author-reviewer discussion period is coming to the end, please let us know if you have any further concerns or questions; we will be happy to address them.
> >
> > Best,
> >
> > Authors

---

> ### Author Response · Authors · 2023-11-22
> **Are there any remaining concerns?**
>
> Dear Reviewer wHfz,
>
> We would like to thank you again for your detailed reviews and suggested experiments to improve this work.
>
> We have conducted **all** required experiments and the results are in the above reply. We hope that our reply has satisfactorily addressed your concerns.
>
> If there is any additional explanation or experiments that can save the reviewer’s time to understand our paper and clarify the concerns, we will be more than happy to do so.
>
> Best,
>
> Authors

---

### Author Response · Authors · 2023-11-22
**General Reply to Reviewers and AC**

Dear Reviewers and AC,

We sincerely thank all the reviewers and ACs for taking time on our submission. We are very grateful for their constructive and valuable comments.

We have carefully responded to every raised concern and hope that our rebuttal satisfactorily addressed them. We have also conducted all the suggested experiments, including:
- Added an ablation experiment on four tasks  to show that our proposed PERU works for fewer tasks (for `Reviewer wHfz`)
- Added an experiment: pruning $\theta_t$ v.s. our PERU-FFT in merging fully finetuned models, to show that pruning the task vectors $\theta_t - \theta^\star$ is more effective (for `Reviewer wHfz`)
- Added an experiment on the *Natural* group of *VTAB* dataset, containing seven similar tasks, to show that the results on *Natural* are consistent with results obtained in the submission (for `Reviewer tpTw`)
- Added an experiment on a CNN-based model *ConvNeXt-Base* to show that the results on the CNN-Based model are consistent with results on the ViT-based models obtained in the submission (for `Reviewer tpTw`)
- Added an experiment: $\theta_0 + \text{Approx}({\bf A}\_t)\text{Approx}({\bf B}\_t)^\top$ v.s. our PERU-LoRA $\theta_0 + \text{Approx}({\bf A}\_t{\bf B}\_t^\top)$ for merging LoRA finetuned models, to show that PERU-LoRA is better (for `Reviewer a2T8`)
- Added an experiment combining our PERU-FFT with existing merging models methods (Fisher-Merging, RegMean, TIES-Merging) to show PERU-FFT is general and can be integrated into existing methods to boost performance (for `Reviewer a2T8`)


We sincerely thank the reviewers for their suggested experiments, which further confirm that our PERU-FFT and PERU-LoRA outperform previous merging methods on a variety of datasets and network architectures. All additional experiments are added in the Appendix of the updated paper and the response below.

Please let us know if you have any remaining questions; we are more than happy to address them.

Thanks again for all the effort and time.

Best,

Authors